# Streaming Drag-Oriented Interactive Video Manipulation: Drag Anything, Anytime!

**Junbao Zhou**[1], **Yuan Zhou**[1*], **Kesen Zhao**[1], **Qingshan Xu**[2], **Beier Zhu**[2], **Richang Hong**[3], **Hanwang Zhang**[1]

[1]Nanyang Technological University    [2]University of Science and Technology of China
[3]Hefei University of Technology

{yuan.zhou, hanwang.zhang}@ntu.edu.sg
{qingshan.xu, beier.zhu}@ustc.edu.cn
{JUNBAO001, KESEN002}@e.ntu.edu.sg, hongrc.hfut@gmail.com

Code: https://github.com/junbao-zhou/DragStream

Project page: https://junbao-zhou.github.io/DragStream.github.io/

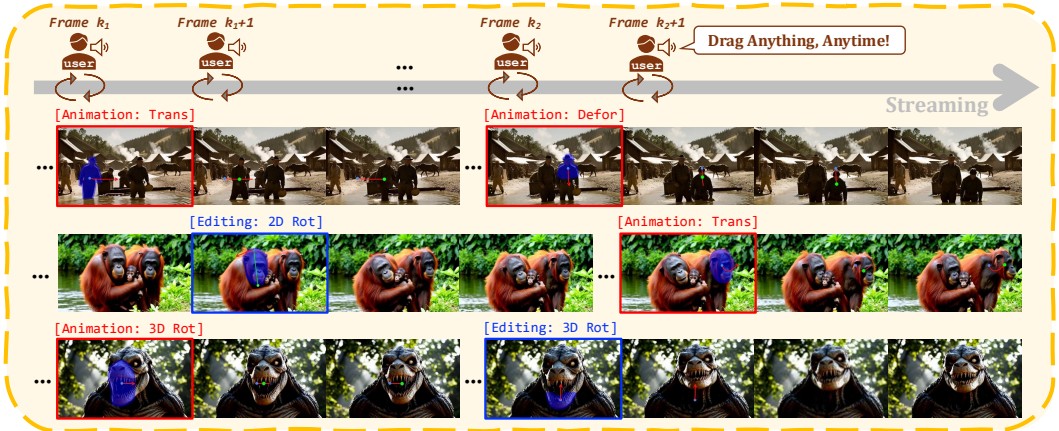

Figure 1: **Examples of our REVEL task.** The streaming video manipulation results shown above—including both `Editing` and `Animation` with drag effects such as object translation ("`Trans`"), deformation ("`Defor`"), and rotation ("`Rot`")—are produced by our **DragStream** method.

## Abstract

Achieving streaming, fine-grained control over the outputs of autoregressive video diffusion models remains challenging, making it difficult to ensure that they consistently align with user expectations. To bridge this gap, we propose **stReaming drag-oriEnted interactiVe vidEo manipuLation (REVEL)**, a new task that enables users to modify generated videos *anytime* on *anything* via fine-grained, interactive drag. Beyond DragVideo and SG-I2V, REVEL unifies drag-style video manipulation as editing and animating video frames with both supporting user-specified translation, deformation, and rotation effects, making drag operations versatile. In resolving REVEL, we observe: *i*) drag-induced perturbations accumulate in latent space, causing severe latent distribution drift that halts the drag process; *ii*) streaming drag is easily disturbed by context frames, thereby yielding visually unnatural outcomes. We thus propose a training-free approach, **DragStream**, comprising: *i*) an adaptive distribution self-rectification strategy that leverages neighboring frames' statistics to effectively constrain the drift of latent embeddings; *ii*) a spatial-frequency selective optimization mechanism, allowing the model to fully exploit contextual information while mitigating its interference via selectively propagating visual cues along generation. Our method can be seamlessly integrated into existing autoregressive video diffusion models, and extensive experiments firmly demonstrate the effectiveness of our DragStream.

---

*Yuan Zhou (yuan.zhou@ntu.edu.sg) is the corresponding author of the paper.

# 1 INTRODUCTION

Video Diffusion Models (VDMs) have shown impressive capabilities in generating photorealistic videos, and their success inspired a broad range of generative applications, including image animation Lei et al. (2025); Hu (2024), text-based video editing Ceylan et al. (2023); Liu et al. (2024), camera-controlled video generation Zheng et al. (2024); He et al. (2024); Bai et al. (2025), etc. With the recent remarkable progress in autoregressive VDMs Yin et al. (2025); Huang et al. (2025) and diffusion acceleration Zhu et al. (2025b); Wang et al. (2025), researchers have been focusing more on achieving controllable video generation in a streaming manner, thereby enabling users to interact with VDMs and alter synthetic videos on the fly. For instance, Kodaira et al. (2023; 2025); Lin et al. (2025) proposed directly finetuning VDMs to support streaming video generation conditioned on text, camera viewpoint, and human pose, whereas Liang et al. (2024) realized training-free, text-guided streaming video translation by introducing a looking-back strategy.

Drag-style operations have become a crucial control signal for VDMs due to their fine-grained nature and user-friendly interactivity Zhou et al. (2025); Wu et al. (2024); Deng et al. (2024); Wang et al. (2024); Namekata et al. (2024); Zhao et al. (2025). However, it remains challenging to realize streaming, fine-grained control over the outputs of VDMs through drag-style operations. To mitigate this dilemma, we propose a new task, **stReaming drag-oriEnted interactiVe vidEo manipuLation (REVEL)**. As shown in Figure 1, REVEL aims to allow users to modify generated videos *at any time* and *on any content* via fine-grained, interactive drag, making generated videos consistently meet users' requirements. We go beyond prior methods, such as DragVideo Deng et al. (2024) and SG-I2V Namekata et al. (2024), by unifying drag-oriented video manipulation as editing and animating video frames, with both supporting user-specified translation, deformation, and rotation effects, thereby making drag operations versatile and establishing a standard paradigm for drag-style video manipulation.

Given the fine-grained nature and high diversity of drag-based video manipulation, solving REVEL is non-trivial. Directly finetuning VDMs to realize REVEL usually incurs expensive training costs—requiring training VDMs on large-scale, fine-grained drag-style data by hundreds or even thousands of H100 GPU hours Yin et al. (2025); Kodaira et al. (2025); Huang et al. (2025)—making it impractical for resource-constrained scenarios. **This observation naturally leads us to ask a key question:** *How can high-quality REVEL be achieved without incurring prohibitive computational costs?*

We propose solving the above question from a training-free perspective in this paper, so as to effectively reduce training expenses. However, we observe that there exist two key challenges: *i*) perturbations induced by drag operations easily accumulate in latent space, thereby causing severe latent distribution drift that totally halts the drag process; *ii*) streaming drag is easily disturbed by context frames, resulting in visually unnatural content. Therefore, we propose a new **DragStream** approach. Specifically, we first design an Adaptive Distribution Self-Rectification (ADSR) strategy that suppresses the distribution drift of latent code by considering statistics from neighboring frames, thereby effectively overcoming drag interruption. We also introduce a Spatial-Frequency Selective Optimization (SFSO) mechanism, which propagates visual cues from preceding video frames selectively in both spatial and frequency domains. As a result, we can fully exploit the information of context frames while relieving their interference. ADSR and SFSO enable our DragStream to achieve high-quality results on REVEL without incurring prohibitive training costs, while allowing it to be seamlessly integrated into existing autoregressive VDMs. Extensive experiments provided in Section 5 and the appendix consistently demonstrate the superiority of our proposed approach.

Here, we summarize the main contributions of this paper:

- We propose **stReaming drag-oriEnted interactiVe vidEo manipuLation (REVEL)**, a new task that enables users to drag *anything anytime* during video generation, thus achieving streaming, fine-grained control over the outputs of VDMs via drag-style operations.

- We identify two key challenges in solving REVEL within a training-free paradigm: *i*) drag-induced perturbations cause severe latent distribution drift and halt the drag process; and *ii*) streaming drag is disturbed by context frames, resulting in visually unnatural outcomes.

- We propose **DragStream**, which incorporates a Spatial-Frequency Selective Optimization (SFSO) mechanism and an Adaptive Distribution Self-Rectification (ADSR) strategy to effectively suppress context interference and mitigate distribution drift in latent code.

- Extensive experiments clearly demonstrate the effectiveness of our approach in addressing REVEL, showing that it achieves high-quality streaming drag-style manipulation, remains training-free, and offers plug-and-play integration with existing autoregressive VDMs.

## 2    RELATED WORK

**Streaming Video Generation.** StreamDiffusion Kodaira et al. (2023), SVDiff Chen et al. (2024), and StreamDiT Kodaira et al. (2025) are recent representative streaming text-guided video generation models, in which VDMs are either trained from scratch or finetuned to enable streaming control via text prompts. Lin et al. (2025) proposed an autoregressive adversarial post-training strategy that enables VDMs to operate as one-step autoregressive generators, supporting conditions on human pose, camera viewpoint, and text. Liang et al. (2024) designed a text-based streaming video translation model by preserving historical information across video frames using a feature bank.

**Drag-Based Video Generation and Editing.** Wu et al. (2024); Wang et al. (2024) proposed finetuning bidirectional VDMs with trajectory conditions, thereby realizing trajectory-guided video generation. Zhang et al. (2025a) proposed unifying text, image, and trajectory conditions into a DiT framework Peebles & Xie (2023), while Geng et al. (2025); Zhang et al. (2025b) further trained VDMs on dense trajectories. Namekata et al. (2024); Qiu et al. (2024); Jain et al. (2024); Deng et al. (2024) resorted to training-free frameworks. Deng et al. (2024) introduced a drag-based latent optimization strategy to realize drag-oriented video editing. Namekata et al. (2024) further considered semantically aligned features Zhu et al. (2025a; 2024; 2025b) during dragging, whereas Qiu et al. (2024) achieved trajectory-guided video generation by imposing guidance on both attention and noise construction.

**REMARK 1.** i) Despite the progress in streaming video generation, current models rarely support highly flexible, fine-grained drag-style operations in a streaming manner—a key challenge our work aims to address. ii) Existing drag-based video generation and editing methods are not tailored for streaming tasks, making them unsuitable for achieving fine-grained, streaming control over the outputs of autoregressive VDMs. iii) Directly finetuning VDMs for realizing streaming drag-style manipulation is computationally expensive, usually requiring training VDMs on large-scale drag-style data by hundreds or even thousands of H100 GPU hours, which is unacceptable for resource-constrained scenarios. Different from finetuning-based methods, our DragStream is training-free and can be seamlessly integrated into existing autoregressive VDMs. iv) Beyond previous works, we unify drag-style video manipulation as editing and animating video frames with both supporting user-specified translation, deformation, and rotation effects, thus making drag operations versatile.

## 3    STREAMING DRAG-ORIENTED INTERACTIVE VIDEO MANIPULATION

We first give the definition of our **stReaming drag-oriEnted interactiVe vidEo manipuLation (REVEL)** task in **Definition 1**. For the summary of the main notations, please refer to Section B.

**Definition 1 (REVEL)** *Let $\Gamma_k$ denote the $k$-th video frame produced by autoregressive VDMs. REVEL aims to enable users to utilize drag-style operations $\boldsymbol{U}_k$ to modify video frames for $\forall k \in \mathbb{Z}^+$ and ensures that subsequently nearby frames are consistent to $\Gamma_k$, so as to realize streaming, fine-grained control over outputs of VDMs and make generated videos always meet users' requirements.*

We argue that there exist a major limitation in current drag-based video manipulation, namely the lack of a unified definition of drag-style manipulation operations. Existing drag-based video editing methods focus on dragging objects in generated videos, with the goal of yielding the effects of *translation*, *deformation*, and *rotation* Deng et al. (2024); also, these methods are generally unable to allow users to animate video frames via dragging. By contrast, trajectory-guided video generation models are designed to generate video clips by moving objects along trajectories, with their motion rendered by VDMs; however, they are not flexible enough to specifically allow users to determine the type of drag operations, e.g., deforming object shape, translating objects, or rotating them around a center point Namekata et al. (2024); Zhang et al. (2025a). Since both of these settings are incomplete, we propose unifying drag-style video manipulation operations in **Proposition 1**.

**Proposition 1 (Unifying Drag-Style Video Manipulation Operations)** *We unify drag-style video manipulation as enabling users to perform editing and animation on video frames via drag-style operations, with both supporting user-specified translation, deformation, and 2D/3D rotation effects.*

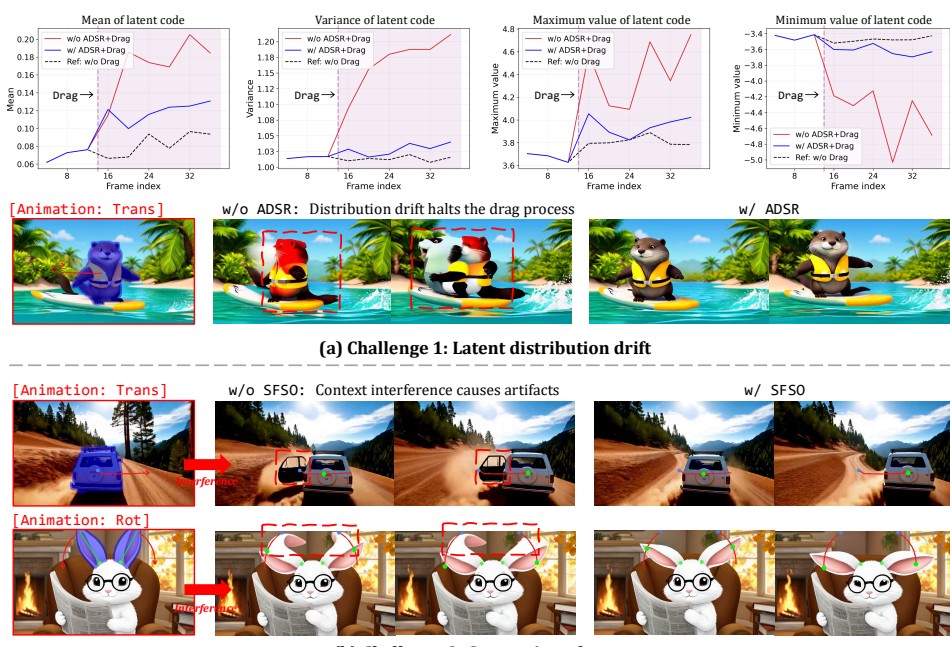

Figure 2: **Examples of Challenge 1 and Challenge 2.**

*Here, editing refers to directly modifying the content of generated video frames, whereas animation represents generating a video clip from an existing frame according to user-given drag instructions.*

**REMARK 2.** Here, we clarify how our REVEL task differs from prior works on drag-based video editing and generation. DragVideo Deng et al. (2024) is a recent typical drag-based video editing approach. Different from our REVEL, it only supports drag-based editing and does not allow users to animate video frames. Moreover, DragVideo does not support the 2D object rotation operation. SG-I2V Namekata et al. (2024) and Tora Zhang et al. (2025a) are two typical trajectory-guided video generation approaches. Both of them focus solely on animating images by moving objects along trajectories with VDM-rendered motion, without allowing users to flexibly achieve more fine-grained drag-style effects, such as editing object shape or rotating objects around a center point by a specific angle. Also DragNeXt Zhou et al. (2025) does not support streaming-style editing. Most importantly, these methods are all incapable of achievinbg drag-oriented video editing and animation in a streaming manner.

We propose addressing REVEL from a training-free perspective, and identify that there exist two key challenges, summarized in **Challenge 1** and **Challenge 2**, respectively.

**Challenge 1 (Latent Distribution Drift)** *Perturbations induced by drag-style operations easily accumulate in the latent space of autoregressive VDMs, which leads to severe distribution drift of latent code and thus interrupts the drag process.*

We show **Challenge 1** in Figure 2 (a). The figure shows that the mean and variance of latent embeddings change significantly once drag operations are applied, while the maximum and minimum values exhibit obvious fluctuations. This instability drives the latent embeddings ("w/o ADSR+drag") to drift away from the original distribution ("Ref: w/o Drag"), thereby disrupting the drag process. We find that latent distribution drift may cause undesirable change of object attributes, such as color and category, as shown in the second row of Figure 2 (a). The use of our ADSR strategy ("w/ ADSR+Drag") can effectively suppress the distribution drift. We will introduce it in Section 4.2.2.

**Challenge 2 (Context Interference)** *Streaming drag is easily disturbed by context frames, misleading VDMs to produce visually unnatural content and thus substantially degrading video quality.*

We show **Challenge 2** in Figure 2 (b). The results in Figure 2 (b) clearly indicate that visual cues from previous frames may mislead the subsequent generation severely, e.g., the features around the handle points spuriously guide the model to produce duplicated ears on the rabbit and artifacts on

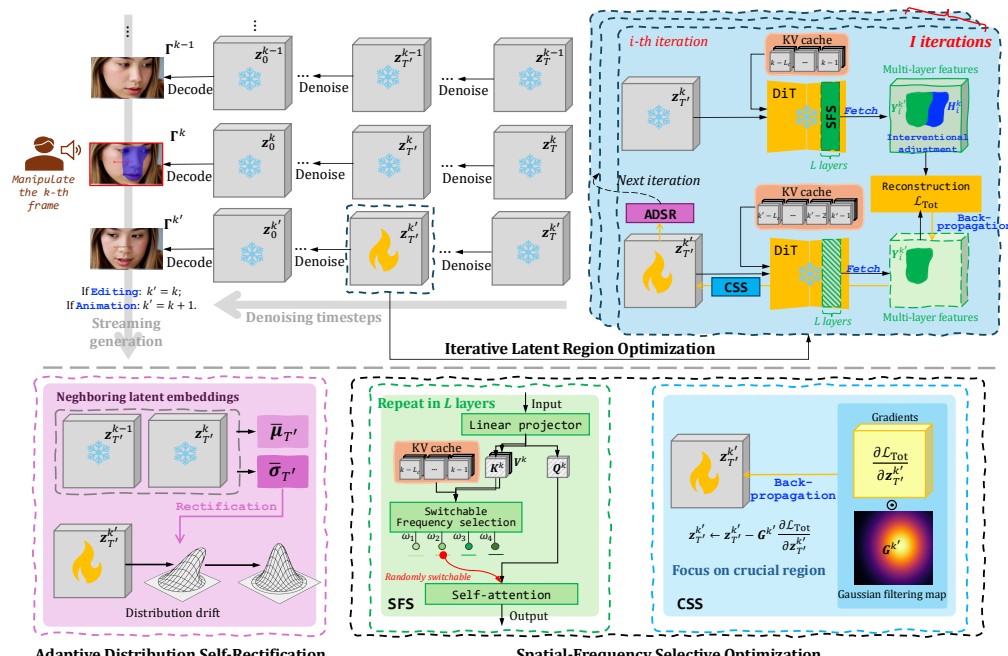

Figure 3: **Schematic illustration of our DragStream,** where an Adaptive Distribution Self-Rectification (ADSR) strategy and a Spatial-Frequency Selective Optimization (SFSO) mechanism are designed to suppress latent distribution drift and context interference, respectively.

the car ("w/o SFSO"), which obviously lowers the quality of generated videos. We will introduce how to overcome context interference by using our SFSO strategy in Section 4.2.3.

## 4 METHODOLOGY

### 4.1 PRELIMINARIES

**Autoregressive Video Diffusion Models.** Autoregressive VDMs refer to a hybrid generative framework that integrates diffusion models with chain-rule decomposition, i.e., $\mathbb{P}(\mathbf{\Gamma}^{1:k}) = \prod_{i=1}^{k} \mathbb{P}(\mathbf{\Gamma}^i \mid \{\mathbf{\Gamma}^j\}_{j=0:i-1})$, where $\mathbb{P}(\mathbf{\Gamma}^i \mid \{\mathbf{\Gamma}^j\}_{j=0:i-1})$ is modeled by iteratively denoising a Gaussian noise latent code $\mathbf{z}_T^i \in \mathcal{N}(\mathbf{0}, \mathbf{I})$ to a clean latent code $\mathbf{z}_0^i$ conditioned on proceeding frames $\{\mathbf{\Gamma}^j\}_{j=0:i-1}$. The KV caching strategy Zhou et al. (2024); Huang et al. (2025); Yin et al. (2025) is often employed during inference to further reduce computations, thereby accelerating autoregressive generation.

**Drag-Style Operation Formats.** Following Zhou et al. (2025), we use $\mathbf{U}^k = \{\mathbf{E}^k, \mathbf{C}^k\}$ to represent drag operations for $\mathbf{\Gamma}^k$, where $\mathbf{E}^k = \{\mathbf{H}_i^k\}_{i=1:n}$ indicates a set of user-specified handle regions that require to be dragged, and $\mathbf{C}^k = \{\eta^k, \zeta_i^k, \mathbf{O}_i^k\}_{i=1:n}$ represents the corresponding drag instructions. The indictor $\eta^k =$ Editing or Animation determines whether the video frame $\mathbf{\Gamma}^k$ is to be edited or animated, whereas $\zeta_i^k$ indicates the type of each drag operation. For animation, $\mathbf{O}_i^k = \{\mathbf{h}_i^k, \{\mathbf{p}_i^{k'}\}_{k'=k+1:k+m}, \mathbf{c}_i^k\}$, if $\zeta_i^k =$ Rotation; otherwise, $\mathbf{O}_i^k = \{\mathbf{h}_i^k, \{\mathbf{p}_i^{k'}\}_{k'=k+1:k+m}\}$. Here, $\mathbf{h}_i^k$ represents a handle point, $\{\mathbf{p}_i^{k'}\}_{k'=k+1:k+m}$ represents $m$ discrete target points sampled along a drag trajectory, assigned to subsequent $m$ video frames, and $\mathbf{c}_i^k$ denotes a rotation center of the handle region $\mathbf{H}_i^k$. For editing, $\mathbf{O}_i^k = \{\mathbf{h}_i^k, \mathbf{p}_i^k, \mathbf{c}_i^k\}$, if $\zeta_i^k =$ Rotation; otherwise, $\mathbf{O}_i^k = \{\mathbf{h}_i^k, \mathbf{p}_i^k\}$. Here, each drag operation considers only one target point, since the editing task ignores intermediate drag states. Also, a binary mask $\mathbf{M}^k$ is utilized to specify the non-editable region of the frame $\mathbf{\Gamma}^k$.

### 4.2 DRAGSTREAM: DRAG ANYTHING, ANYTIME IN A TRAINING-FREE PARADIGM

#### 4.2.1 OVERALL PIPELINE

We first introduce the overall pipeline of our DragStream. Suppose that users observe the video frame $\mathbf{\Gamma}^k$ during streaming generation and intend to manipulate $\mathbf{\Gamma}^k$ by giving the instructions $\mathbf{U}^k = \{\mathbf{E}^k, \mathbf{C}^k\}$, where $\mathbf{E}^k = \{\mathbf{H}_i^k\}_{i=1:n}$ indicates handle regions, and $\mathbf{C}^k = \{\eta^k, \zeta_i^k, \mathbf{O}_i^k\}_{i=1:n}$

denotes the corresponding drag instructions. We use $\mathbf{\Gamma}^{k'}$ to represent a video frame produced during dragging, where $k' = k$ if $\eta^k = \texttt{Editing}$; otherwise, $k' > k$ since new frames are animated during $\texttt{Animation}$.

We take the handle region $\boldsymbol{H}_i^k$ as an example to illustrate our method. As exhibited in Figure 3, we first denoise $\boldsymbol{z}_T^{k'}$ to $\boldsymbol{z}_{T'}^{k'}$, and extract the features $\mathcal{F}(\boldsymbol{z}_{T'}^{k'})$ by concatenating features from the multiple layers of the DiT denoiser $\epsilon_{\boldsymbol{\Theta}}(\cdot | \{\boldsymbol{K}^i, \boldsymbol{V}^i\}_{i=0:k'-1})$, where $\{\boldsymbol{K}^i, \boldsymbol{V}^i\}_{i=0:k'-1}$ are the cached keys and values of context frames. We then estimate the position of the handle region $\boldsymbol{H}_i^k$ after being dragged within the features $\mathcal{F}(\boldsymbol{z}_{T'}^{k'})$ according to the user-given drag instruction:

$$\boldsymbol{Y}_i^{k'}, \boldsymbol{\Pi}_{\boldsymbol{H}_i^k \to \boldsymbol{Y}_i^{k'}} = \mathcal{G}(k', \boldsymbol{H}_i^k, \eta^k, \zeta_i^k, \boldsymbol{O}_i^k), \tag{1}$$

$$s.t., \mathcal{G}(k', \boldsymbol{H}_i^k, \eta^k, \zeta_i^k, \boldsymbol{O}_i^k) = \begin{cases} \texttt{Rot}(\boldsymbol{H}_i^k, \boldsymbol{c}_i^k, \theta = \angle \boldsymbol{p}_i^{k'} \boldsymbol{c}_i^k \boldsymbol{p}_i^k), & \text{if } \zeta_i^k = \texttt{Rotation} \\ \texttt{Trans}(\boldsymbol{H}_i^k, \boldsymbol{\vartheta} = \boldsymbol{p}_i^{k'} - \boldsymbol{p}_i^k), & \text{else.} \end{cases}$$

Here, $\texttt{Rot}(\boldsymbol{H}_i^k, \boldsymbol{c}_i^k, \theta)$ denotes rotating the handle region $\boldsymbol{H}_i^k$ around the center point $\boldsymbol{c}_i^k$ by an angle $\theta$, and $\texttt{Trans}(\boldsymbol{H}_i^k, \boldsymbol{\vartheta})$ indicates translating $\boldsymbol{H}_i^k$ by an offset $\boldsymbol{\vartheta}$. $\boldsymbol{Y}_i^{k'}$ is a binary mask that indicates the target position of $\boldsymbol{H}_i^k$ in the extracted features $\mathcal{F}(\boldsymbol{z}_{T'}^{k'})$, and $\boldsymbol{\Pi}_{\boldsymbol{H}_i^k \to \boldsymbol{Y}_i^{k'}}$ is the coordinate mapping from $\boldsymbol{H}_i^k$ to $\boldsymbol{Y}_i^{k'}$. Finally, the latent code $\boldsymbol{z}_{T'}^{k'}$ is iteratively optimized. In each iteration, the features of $\boldsymbol{z}_{T'}^k$ are also extracted and detached as reference features, $\mathcal{F}_{\texttt{ref}}(\boldsymbol{z}_{T'}^k) = \mathcal{F}(\boldsymbol{z}_{T'}^k).\texttt{detach()}$. Moreover, we interventionally adjust the reference features according to the coordinate mapping, $\mathcal{F}_{\texttt{ref}}(\boldsymbol{z}_{T'}^k)[\boldsymbol{\Pi}_{\boldsymbol{H}_i^k \to \boldsymbol{Y}_i^{k'}}]$, thereby perturbing the original latent code and transforming the handle region features to the target position $\boldsymbol{Y}_i^{k'}$. The latent code $\boldsymbol{z}_{T'}^{k'}$ of the new frame $\mathbf{\Gamma}^{k'}$ can be updated by reconstructing the features from the original handle region at the target position of $\mathcal{F}(\boldsymbol{z}_{T'}^{k'})$

$$\boldsymbol{z}_{T'}^{k'} \longleftarrow \boldsymbol{z}_{T'}^{k'} - \frac{\partial \mathcal{L}_{\texttt{Tot}}}{\partial \boldsymbol{z}_{T'}^{k'}}, \tag{2}$$

where

$$\mathcal{L}_{\texttt{Tot}} = \underbrace{\|\mathcal{F}(\boldsymbol{z}_{T'}^{k'}) * \boldsymbol{Y}_i^{k'} - \mathcal{F}_{\texttt{ref}}(\boldsymbol{z}_{T'}^k)[\boldsymbol{\Pi}_{\boldsymbol{H}_i^k \to \boldsymbol{Y}_i^{k'}}] * \boldsymbol{Y}_i^{k'}\|_1}_{\mathcal{L}_{\texttt{Rec}}} + \underbrace{\|\mathcal{F}(\boldsymbol{z}_{T'}^{k'}) * \boldsymbol{M}^{k'} - \mathcal{F}_{\texttt{init}}(\boldsymbol{z}_{T'}^{k'}) * \boldsymbol{M}^{k'}\|_1}_{\mathcal{L}_{\texttt{Cst}}}.$$

$$\tag{3}$$

Here, $\mathcal{L}_{\texttt{Rec}}$ denotes a reconstruction loss, and $\mathcal{L}_{\texttt{Cst}}$ represents a constraint term that ensures the consistency of the non-editable region $\boldsymbol{M}^{k'}$ of $\mathbf{\Gamma}^{k'}$. $\mathcal{F}_{\texttt{init}}(\boldsymbol{z}_{T'}^{k'}) = \mathcal{F}(\boldsymbol{z}_{T'}^{k'}).\texttt{detach()}$ indicates the initial features of $\boldsymbol{z}_{T'}^{k'}$ before conducting iterative latent region optimization. Our ADSR and SFSO strategies are employed during the above iterative latent region optimization process to overcome **Challenge 1** and **Challenge 2**, which are detailed in Section 4.2.2 and Section 4.2.3, respectively.

**REMARK 2.** If $\eta^k = \texttt{Animation}$, then $k' > k$, which represents a cross-frame optimization paradigm, i.e., using the perturbed features $\mathcal{F}_{\texttt{ref}}(\boldsymbol{z}_{T'}^k)[\boldsymbol{\Pi}_{\boldsymbol{H}_i^k \to \boldsymbol{Y}_i^{k'}}]$ to guide the denoising process of $\boldsymbol{z}_T^{k'}$ of the new frame $\mathbf{\Gamma}^{k'}$. If $\eta^k = \texttt{Editing}$, $k' = k$, which can be seen as self-guided optimization, i.e., using the detached features $\mathcal{F}_{\texttt{ref}}(\boldsymbol{z}_{T'}^k)[\boldsymbol{\Pi}_{\boldsymbol{H}_i^k \to \boldsymbol{Y}_i^{k'}}]$ of $\mathbf{\Gamma}^k$ to guide the re-denoising of $\boldsymbol{z}_T^k$.

### 4.2.2 ADAPTIVE DISTRIBUTION SELF-RECTIFICATION

We propose a simple-yet-effective strategy, Adaptive Distribution Self-Rectification (ADSR), to address the latent distribution drift issue caused by cumulative perturbations—**Challenge 1**—as provided in **Proposition 2**.

**Proposition 2 (Adaptive Distribution Self-Rectification)** *Suppose users apply drag-style operations to the frame $\mathbf{\Gamma}_k$. The statistics $\bar{\boldsymbol{\mu}}_{T'}$ and $\bar{\boldsymbol{\sigma}}_{T'}$ of the preceding neighboring latent embeddings $\{\boldsymbol{z}_{T'}^i\}_{i=k'-L_n-1:k'-1}$ of $\mathbf{\Gamma}_k$ are recorded, where $\bar{\boldsymbol{\mu}}_{T'}$ and $\bar{\boldsymbol{\sigma}}_{T'}$ are the mean and standard deviation. We propose using $\bar{\boldsymbol{\mu}}_{T'}$ and $\bar{\boldsymbol{\sigma}}_{T'}$ to rectify the distribution of $\boldsymbol{z}_{T'}^{k'}$ after each optimization iteration:*

$$\hat{\boldsymbol{z}}_{T'}^{k'} = \frac{\texttt{Iter\_optim}(\boldsymbol{z}_{T'}^{k'}, \boldsymbol{U}^k) - \boldsymbol{\mu}_{T'}^k}{\boldsymbol{\sigma}_{T'}^{k'}} * \bar{\boldsymbol{\sigma}}_{T'} + \bar{\boldsymbol{\mu}}_{T'}, \tag{4}$$

where $\mathtt{Iter\_optim}(\cdot)$ denotes an iteration of the latent optimization, and $\bar{\boldsymbol{\mu}}_{T'}/\boldsymbol{\mu}_{T'}^{k'}$ and $\bar{\boldsymbol{\sigma}}_{T'}/\boldsymbol{\sigma}_{T'}^{k'}$ denotes the mean and standard deviation of $\{\boldsymbol{z}_{T'}^{i}\}_{i=k'-L_n-1:k'-1}/\boldsymbol{z}_{T'}^{k'}$. As exemplified in Figure 2 (a), our ADSR can effectively suppress the distribution drift of latent embeddings, while significantly improving video quality and preventing undesired changes in object attributes during dragging. This aligns with the findings provided in Figure 7, showing that ADSR consistently improves model performance across the evaluation metrics ObjMC, DAI, FVD, and FID. For more details, please refer to Section 5.

### 4.2.3 SPATIAL-FREQUENCY SELECTIVE OPTIMIZATION

We design a Spatial-Frequency Selective Optimization (SFSO) mechanism to overcome **Challenge 2**. It fully exploits the information of context frames while relieving their interference via conducting information selection in both frequency and spatial domains during iterative latent region optimization.

High-frequency information—though capturing finer visual information—tends to mislead VDMs to produce unnatural results, as it carries more noise perturbations Fan et al. (2019); Li et al. (2020); by contrast, low-frequency information—while more robust—lacks sufficient fine-grained visual details. We argue that it is crucial to harness the strengths of both high- and low-frequency information while alleviating their inherent limitations during the drag-oriented optimization process. We therefore propose a Switchable Frequency-domain Selection (SFS) strategy in **Proposition 3**.

**Proposition 3 (Switchable Frequency-domain Selection)** *Let $\{l_i\}_{i=1:L}$ represent the layers of the DiT denoiser that are used to construct reference features, and let $\boldsymbol{X}_{l_i}^{k}$ denote the input features of the layer $l_i$. SFS is applied to the self-attention of the layer $\{l_i\}_{i=1:L}$ to build reference features with switchable frequency components in each iteration of the latent region optimization process:*

$$\boldsymbol{Q}_{l_i}^{k}, \boldsymbol{K}_{l_i}^{k}, \boldsymbol{V}_{l_i}^{k} = \mathtt{Linear\_projector}(\boldsymbol{X}_{l_i}^{k}), \tag{5}$$

$$\bar{\boldsymbol{K}}_{l_i}^{k} = \mathtt{Concat}(\{\boldsymbol{K}_{l_i}^{j}\}_{j=0:k-1}, \boldsymbol{K}_{l_i}^{k}), \bar{\boldsymbol{V}}_{l_i}^{k} = \mathtt{Concat}(\{\boldsymbol{V}_{l_i}^{j}\}_{j=0:k-1}, \boldsymbol{V}_{l_i}^{k}), \tag{6}$$

$$\{\bar{\boldsymbol{K}}_{l_i}^{k}, \bar{\boldsymbol{V}}_{l_i}^{k}\} = \mathtt{IFFT}(\mathtt{Butterw}(\mathtt{FFT}(\{\bar{\boldsymbol{K}}_{l_i}^{k}, \bar{\boldsymbol{V}}_{l_i}^{k}\}), \omega = \mathtt{Random}(\omega_1, ..., \omega_N))), \tag{7}$$

$$\bar{\boldsymbol{X}}_{l_i}^{k} = \mathtt{self-attention}(\boldsymbol{Q}_{l_i}^{k}, \bar{\boldsymbol{K}}_{l_i}^{k}, \bar{\boldsymbol{V}}_{l_i}^{k}). \tag{8}$$

*Here, $\{\boldsymbol{K}_{l_i}^{j}\}_{j=0:k-1}$ and $\{\boldsymbol{V}_{l_i}^{j}\}_{j=0:k-1}$ denote cached keys and values, $\bar{\boldsymbol{X}}_{l_i}^{k}$ denotes the extracted reference features of the layer $l_i$, $\mathtt{Butterw}(\cdot \mid \omega)$ represents the Butterworth filter with the cutoff frequency $\omega$ randomly selected from $\{\omega_i\}_{i=1:N}$, and $\mathtt{FFT}(\cdot)$ and $\mathtt{IFFT}(\cdot)$ represent the 2D Fourier transform and 2D inverse Fourier transform.*

By using SFS strategy, in each iteration, the information of different frequencies can be propagated to the latent embeddings $\boldsymbol{z}_{T'}^{k'}$ of $\boldsymbol{\Gamma}^{k'}$ by the reconstruction loss $\mathcal{L}_{\mathtt{Rec}}$, thus fully exploiting information from context frames, while preventing high-frequency information from dominating the drag process and inducing artifacts in generated frames.

In **Proposition 4**, we also design a Criticality-driven Spatial-domain Selection (CSS) strategy to prevent over-optimization of the background within editable region, which is beneficial for further reducing unnatural content.

**Proposition 4 (Criticality-driven Spatial-domain Selection)** *We selectively back-propagate gradients in spatial domain, avoiding the drag process undesirably affecting the background:*

$$\boldsymbol{z}_{T'}^{k'} \longleftarrow \boldsymbol{z}_{T'}^{k'} - \boldsymbol{G}^{k'} \frac{\partial \mathcal{L}_{\mathtt{Tot}}}{\partial \boldsymbol{z}_{T'}^{k'}} \tag{9}$$

where $\boldsymbol{G}^{k'}$ is a Gaussian filtering map that decays w.r.t. the distance to the center point $(x_c, y_c)$ of the edited region

$$\boldsymbol{G}^{k'}[x, y] = \exp\left[-\left(\frac{(x-x_c)^2}{2\sigma_x^2} + \frac{(y-y_c)^2}{2\sigma_y^2}\right)\right], \ s.t., \ \sigma_x = \frac{W}{2} * \alpha \text{ and } \sigma_y = \frac{H}{2} * \alpha, \tag{10}$$

$W$ and $H$ are the width and height of the handle region's minimum bounding rectangle, and $\alpha$ is a hyperparameter scaling the spread of the Gaussian and set as $1$. The use of SFS and CSS can further improve video quality, which is demonstrated by experiments given in the main paper and appendix.

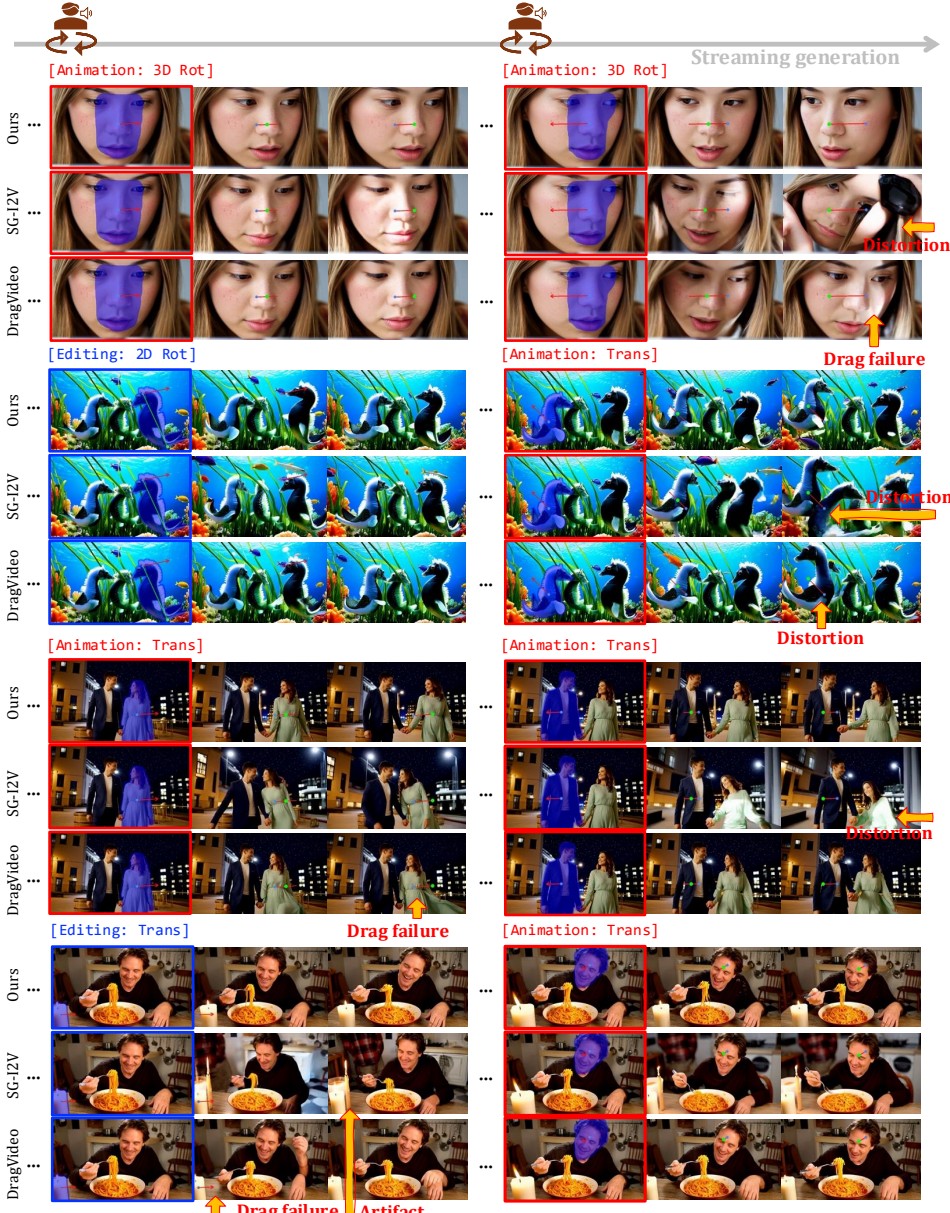

Figure 4: **Visualization results achieved by our DragStream on REVEL.** Note that `Editing` produces only one video frame, but we insert an extra subsequent frame to maintain layout consistency with `Animation`.

## 5 EXPERIMENTS

Since REVEL is a new task, no existing approaches have been specifically designed to tackle it. We adapt two training-free methods, SG-I2V Namekata et al. (2024) and DragVideo Deng et al. (2024), to the REVEL task for comparison. Please refer to Section C of the appendix for details about our experimental setup, including implementation details, evaluation metrics, and compared baselines.

### 5.1 MAIN RESULTS

**Visualization Results.** The visualization results achieved by our method are shown in Figure 4. Compared to SG-I2V and DragVideo, our DragStream produces obviously more natural and higher-quality streaming drag-style video manipulation results. For instance, it better preserves object appearance and structure, while exhibiting fewer visual distortions, artifacts, and drag failures. These results validate the effectiveness of our method in addressing the REVEL task. More visualization results achieved by our DragStream are provide in the appendix; for details, please refer to Section E.

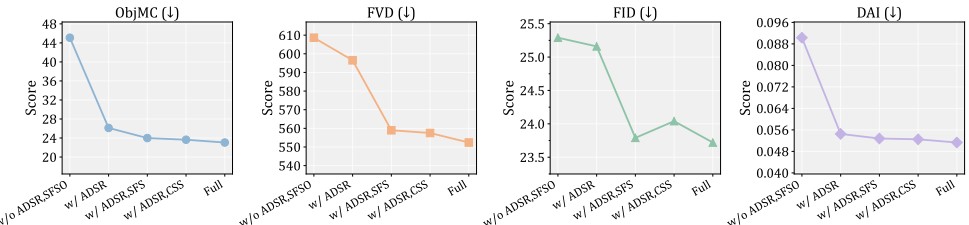

Figure 5: **Quantitative performance achieved by our method in terms of ObjMC, FVD, FID, and DAI.** "↓" indicates that lower values correspond to better performance.

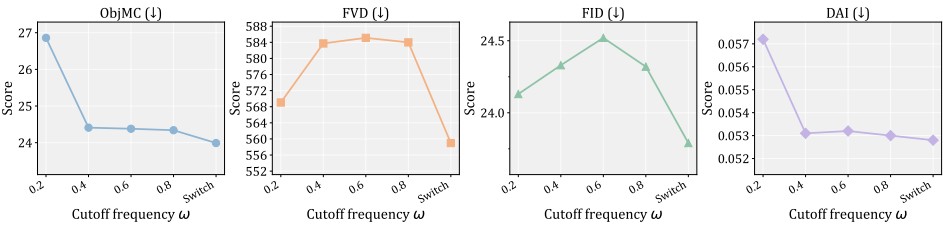

Figure 6: **Ablation study on the key components of our DragStream.**

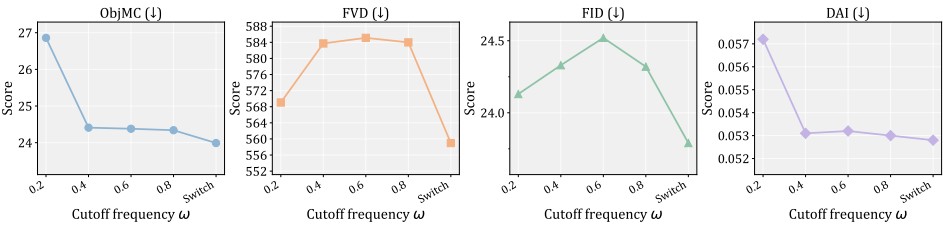

Figure 7: **Analysis on the influence of the cutoff frequency $\omega$.** "Switch" represents frequencies are switchable during the latent region optimization.

**Quantitative Performance.** The quantitative results in Figure 5 demonstrate that our DragStream consistently outperforms SG-I2V and DragVideo again. On one hand, the lowest FID and FVD scores indicate that our DragStream achieves higher video quality than SG-I2V and DragVideo. On the other hand, achieving the best ObjMC and DAI scores demonstrates that our DragStream approach realizes more precise object dragging, aligned with the findings shown in Figure 4.

## 5.2 Analysis

**Ablation Study.** In Figure 6, we conduct ablation study to investigate the influence of each component. The results indicate the full method achieves the best performance. Discarding SFSO ("w/ ADSR") leads to significant performance degradation, while further removing ADSR ("w/o ADSR, SFSO") results in an even greater decline. These results demonstrate the importance of the ADSR strategy and the SFSO mechanism. Similarly, using the full SFSO is better than using CSS or SFS alone. We also analyze the influence of the cutoff frequency in Figure 7. We can see that both small and large cutoff frequencies lead to performance drops. By contrast, our switchable frequency selection strategy achieves the best performance, as it fully exploits contextual information while mitigating high-frequency interference by preventing them from dominating the drag process.

**Runtime Analysis.** Table 1 exhibits the runtime analysis of our DragStream approach. Our DragStream is based on an iterative optimization scheme. In the table, we investigate the influence of the iteration number $I$. We find that setting $I = 4$ already achieves satisfactory performance, achieving 23.05 ObjMC and 0.051 DAI, while incurring only 0.13s of additional runtime per frame compared with the baseline without DragStream (i.e., $I = 0$). Decreasing the itera-

Table 1: **Runtime analysis of our DragStream approach.** In the table, "RF" denotes runtime per frame, and $I$ indicates the number of iterations of drag-oriented latent optimization.

| Experiments | RF | ObjMC (↓) | DAI (↓) |
|:---:|:---:|:---:|:---:|
| $I = 0$ | **0.17s** | 90.39 | 0.133 |
| $I = 2$ | 0.24s | 27.67 | 0.054 |
| $I = 3$ | 0.27s | 24.55 | 0.053 |
| $I = 4$ **(Ours)** | 0.30s | **23.05** | **0.051** |

tion number—such as $I = 2$ or $3$—can further improve execution speed, while still maintaining acceptable drag-based manipulation performance, with ObjMC and DAI clearly outperforming those of the baseline (i.e., $I = 0$). All the experiments on the table are conducted on a NVIDIA H20 GPU.

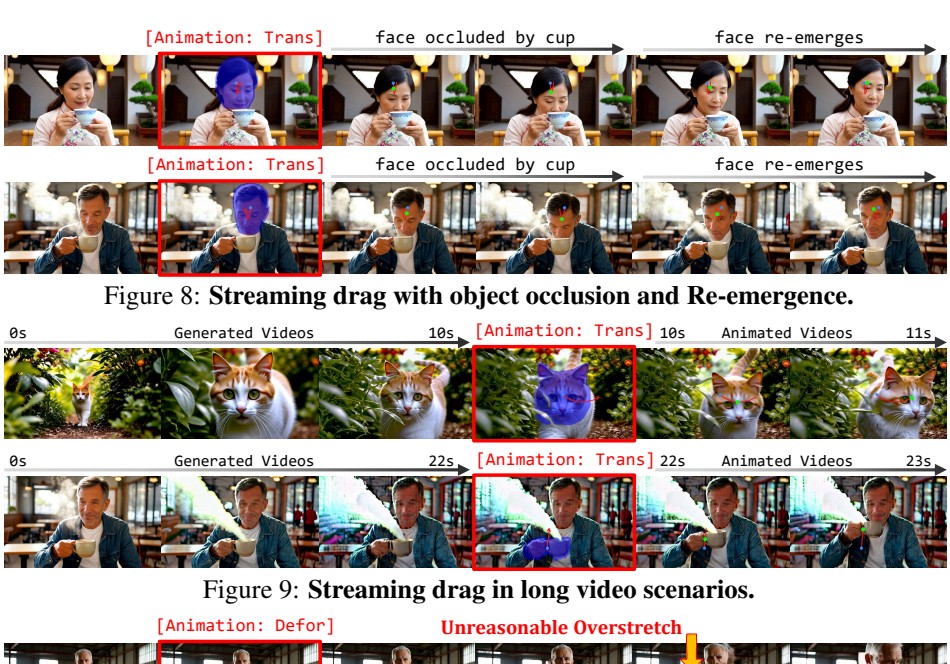

Figure 8: **Streaming drag with object occlusion and Re-emergence.**

Figure 9: **Streaming drag in long video scenarios.**

Figure 10: **Failure cases under unreasonable and physically implausible conditions.**

### 5.3 COMPLEX STREAMING MANIPULATION

**Occlusion and Re-emergence.** In Figure 8, we also study our DragStream in the scenario of object occlusion and subsequent re-emergence. We find that our approach shows promising performance in this scenario and produce smooth video results. This is because VDMs are trained on massive amounts of data and thereby learns rich prior knowledge about object occlusion and scene transition.

**Streaming Drag in Long Video Generation.** In Figure 9, we study the use of our DragStream for achieving streaming drag in long video generation. As shown in the figure, despite that accumulated errors remain a challenging issue for current autoregressive VDMs, our method can still effectively realize drag-based manipulation. For more results, please refer to Section P of our appendix.

### 5.4 FAILURE CASES

We observe a failure case of our method. As shown in Figure 10, our method fails to realize high-quality manipulation under highly unreasonable and physically implausible conditions, as such manipulation instructions severely conflict with prior knowledge learned by VDMs in large-scale data.

## 6 CONCLUSION

We propose **stReaming drag-oriEnted interactiVe vidEo manipuLation (REVEL)**, a new task that aims to allow users to achieve streaming, drag-style control over the outputs of autoregressive VDMs. To solve REVEL, we propose a training-free approach, **DragStream**, which employs an Adaptive Distribution Self-Rectification (ADSR) strategy and design a Spatial-Frequency Selective Optimization (SFSO) mechanism. ADSR effectively constrains the drift of latent embeddings by leveraging neighboring frames' statistics, while SFSO fully exploits contextual information while mitigating its interference via selectively propagating visual cues along generation in spatial and frequency domains. These two strategies enable our method to achieve superior performance on REVEL and allow seamless integration into existing autoregressive VDMs. We hope this work will inspire more excellent solutions to address the streaming drag-style video manipulation problem.

ACKNOWLEDGMENTS

This research is supported by the National Research Foundation, Singapore under its AI Singapore Programme (AISG Award No: AISG3-RP-2022-030) and NRF-NRFI10-2024-0004.

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

CONTENTS

## A    USE OF LLMs

LLMs were only used to provide minor writing assistance in preparing the manuscript, such as grammar polishing and readability improvement. No parts of the methodology, experimental design, analysis, or results were generated by LLMs. The all ideas, experiments, and conclusions are entirely completed and drawn by the authors of this paper.

## B    SUMMARY OF MAIN NOTATIONS

In Table 2, we provide a summary for the main notations used in this paper.

Table 2: **Summary of main notations.**

| Notions | Descriptions |
|---|---|
| $\boldsymbol{\Gamma}^k$ | The $k$-th video frame generated by VDMs. |
| $\boldsymbol{z}_T^k$ | The latent embeddings of $\boldsymbol{\Gamma}^k$ at the denoising timestep $T$. |
| $\boldsymbol{U}^k = \{\boldsymbol{E}^k, \boldsymbol{C}^k\}$ | The user-specified drag-style operations for the frame $\boldsymbol{\Gamma}^k$. |
| $\boldsymbol{E}^k = \{\boldsymbol{H}_i^k\}_{i=1:n}$ | The set of user-specified handle regions for the frame $\boldsymbol{\Gamma}^k$. |
| $\boldsymbol{C}^k = \{\eta^k, \zeta_i^k, \boldsymbol{O}_i^k\}_{i=1:n}$ | The corresponding drag instructions for the handle region $\boldsymbol{E}^k$. |
| $\boldsymbol{H}_i^k$ | The binary mask that indicates the $i$-th user-specified handle region of the frame $\boldsymbol{\Gamma}^k$. |
| $\boldsymbol{Y}_i^{k'}$ | The binary mask indicates the target position of $\boldsymbol{H}_i^k$ that is to be dragged in $\boldsymbol{\Gamma}^{k'}$. |
| $\boldsymbol{\Pi}_{\boldsymbol{H}_i^k \to \boldsymbol{Y}_i^{k'}}$ | The coordinate mapping from the handle region $\boldsymbol{H}_i^k$ to the target position $\boldsymbol{Y}_i^{k'}$ |
| $\eta^k$ | The indictor determines whether the frame $\boldsymbol{\Gamma}^k$ is to be edited or animated. |
| $\zeta_i^k$ | The indictor determines the type of drag operations, i.e, translation, deformation, and rotation. |
| $\boldsymbol{O}_i^k$ | The points sampled from the drag trajectory given for the handle region $\boldsymbol{H}_i^k$. |
| $\boldsymbol{M}^k$ | The user-specified non-editable region of the video frame $\boldsymbol{\Gamma}^k$. |
| $\boldsymbol{G}^k$ | The Gaussian filtering map used for the frame $\boldsymbol{\Gamma}^k$ during latent region optimization. |
| $\theta = \angle \boldsymbol{p}_i^{k'} \boldsymbol{c}_i^k \boldsymbol{p}_i^k$ | The angle at the center point $\boldsymbol{c}_i^k$ w.r.t. the trajectory points $\boldsymbol{p}_i^{k'}$ and $\boldsymbol{p}_i^k$. |
| $\vartheta = \boldsymbol{p}_i^{k'} - \boldsymbol{p}_i^k$ | The offset of the sampled trajectory point $\boldsymbol{p}_i^{k'}$ w.r.t. $\boldsymbol{p}_i^k$. |
| $\boldsymbol{\mu}_{T'}^{k'} / \boldsymbol{\sigma}_{T'}^{k'}$ | The mean/standard deviation of the latent embeddings $\boldsymbol{z}_{T'}^{k'}$. |
| $\boldsymbol{Q}^k / \boldsymbol{K}^k / \boldsymbol{V}^k$ | Query/key/value features about the frame $\boldsymbol{\Gamma}^k$. |
| $\mathcal{L}_{\text{Tot}}$ | The object function used in the latent region optimization. |
| $\mathcal{L}_{\text{Rec}}$ | The reconstruction loss used in the latent region optimization. |
| $\mathcal{L}_{\text{Cst}}$ | The constraint term in the latent region optimization. |
| $\epsilon_{\boldsymbol{\Theta}}(\cdot)$ | The denoiser of VDMs. |
| $\mathcal{F}(\cdot)$ | The function that extracts the features of latent code from $\epsilon_{\boldsymbol{\Theta}}(\cdot)$. |

## C    EXPERIMENTAL SETUP

### C.1    IMPLEMENTATION DETAILS

We implement our DragStream in PyTorch and run it on an NVIDIA H20 GPU card. We choose Self-Forcing Huang et al. (2025) as our main base autoregressive VDM with the number of denoising timesteps $T = 4$. We follow SG-I2V to use the AdamW Loshchilov & Hutter (2017) optimizer during latent optimization, with the learning rate set as $4 \times 10^{-2}$. Following Deng et al. (2024), we perform latent region optimization at the denoising timestep $T' = 3$, where the features of latent code are extracted from the $12-15$ layers of the DiT denoiser, the number of iterations is set to $I = 4$ per trajectory point, and the set of cutoff frequencies is set as $\{0.2, 0.4, 0.6, 1\}$. Following Zhang et al. (2025a), we annotate 204 video clips generated by Self-Forcing with diverse drag trajectories and scenes, to serve as a new benchmark for evaluating model performance on our proposed REVEL task, i.e., realizing fine-grained, drag-style control over the outputs of video generation models.

### C.2    EVALUATION METRICS

We evaluate model performance on the REVEL task using four metrics: Fréchet Video Distance (FVD) Unterthiner et al. (2018), Fréchet Inception Distance (FID) Heusel et al. (2017), DAI Zhang et al. (2024), and ObjMC Wu et al. (2024). Since FVD and FID are well-defined metrics, we omit

the explanation for them; for more details, please refer to Unterthiner et al. (2018); Heusel et al. (2017). We provide the details of ObjMC and DAI below.

**ObjMC.** ObjMC Wu et al. (2024) is a metric to evaluate the motion fidelity of the manipulated object in the video. It is calculated as the average distance between the trajectory of the manipulated object in the generated video and the groundtruth trajectory specified by the user. To generate the trajectory of the manipulated object, we utilize Co-Tracker 3 Karaev et al. (2024) to track the points scattered in the original region of the manipulated object, and then compute the average position of these points in each frame to form the trajectory. Lower ObjMC scores indicate that the manipulated object in the generated video closely follows the user's specified trajectory, reflecting better motion fidelity.

**DAI.** DAI Zhang et al. (2024) is a metric for evaluating the quality of drag editing in image and video manipulation. Specifically, DAI calculates the average difference between the latent features of the original handle region and the corresponding manipulated region in the final output. The metric is defined as:

$$\text{DAI} = \frac{1}{N} \sum_{i=1}^{N} \frac{\|z_0^{k'}[\Omega(\boldsymbol{p}_i^{k'}, r)] - z_0^{k}[\Omega(\boldsymbol{p}_i^{k}, r)]\|}{(1 + 2r)^2} \tag{11}$$

where $N$ is the number of trajectory points, while $z_0^k$ and $z_0^{k'}$ are the latent embeddings of the original frame $\boldsymbol{\Gamma}^k$ and the manipulated frame $\boldsymbol{\Gamma}^{k'}$, respectively. $r$ is the radius of the area we calculate DAI and $\Omega(\boldsymbol{p}, r)$ denotes a square area centered at point $\boldsymbol{p}$ with a side length of $2r + 1$. $\boldsymbol{p}_i^k$ and $\boldsymbol{p}_i^{k'}$ are the $i$-th trajectory points in the original frame $\boldsymbol{\Gamma}^k$ and the manipulated frame $\boldsymbol{\Gamma}^{k'}$, respectively. We set $r = 20$ following Zhou et al. (2025), which is suitable for measuring the consistency inside the manipulated region. Lower DAI scores indicate that the manipulated region in the generated image or video closely matches the target region specified by the user, reflecting better drag editing quality.

### C.3 COMPARED BASELINES

We emphasize that our proposed task, **stReaming drag-oriEnted interactiVe vidEo manipuLation (REVEL)**, is entirely new. Thus, to the best of our knowledge, no existing approaches have been specifically designed to address it. For comparison, we adapt two training-free approaches, SG-I2V Namekata et al. (2024) and DragVideo Deng et al. (2024), to our REVEL setting. *Like our DragStream, both SG-I2V and DragVideo follow the latent optimization paradigm; however, they are not equipped with our ADSR and SFSO strategies, designed to address **Challenge 1** and **Challenge 2** in REVEL.* We do not include the finetuning–based methods such as Tora Zhang et al. (2025a) and DragAnything Wu et al. (2024) in our comparisons. First, our DragStream is training-free, making direct comparisons with finetuning–based methods unfair. Second, Tora and DragAnything are not specifically designed to solve our REVEL. Adapting them to enable autoregressive generation or streaming control of VDMs would be non-trivial, as it requires finetuning on large-scale, fine-grained drag-style data by hundreds or even thousands of H100 GPU hours Yin et al. (2025); Kodaira et al. (2025); Huang et al. (2025), which stands in direct contrast to the core philosophy of our work—achieving high-quality REVEL without incurring prohibitive training costs. We leave exploring the REVEL task from a finetuning-based perspective for our future research.

## D  STREAMING VS. NON-STREAMING DRAG-STYLE VIDEO MANIPULATION

We provide a comparison between streaming and non-streaming drag-style video manipulation in Figure 6. A fundamental distinction lies in the different type of used VDMs. In streaming drag-style video manipulation, video frames are generated autoregressively; thus, when an unsatisfactory video frame is observed, users can directly feed drag-style operations to models and modify videos on the fly. In contrast, non-streaming drag-style video manipulation relies on conventional bidirectional VDMs that generate an entire video clip at each time as they are based on modeling bidirectional information across frames, requiring users to regenerate the whole video clip when they find a frame unsatisfactory. *That is why adapting finetuning–based, non-streaming drag-style video manipulation approaches to REVEL is non-trivial as we mentioned in Section C.3: it would require transforming the bidirectional generation paradigm totally into the autoregressive manner, which in*

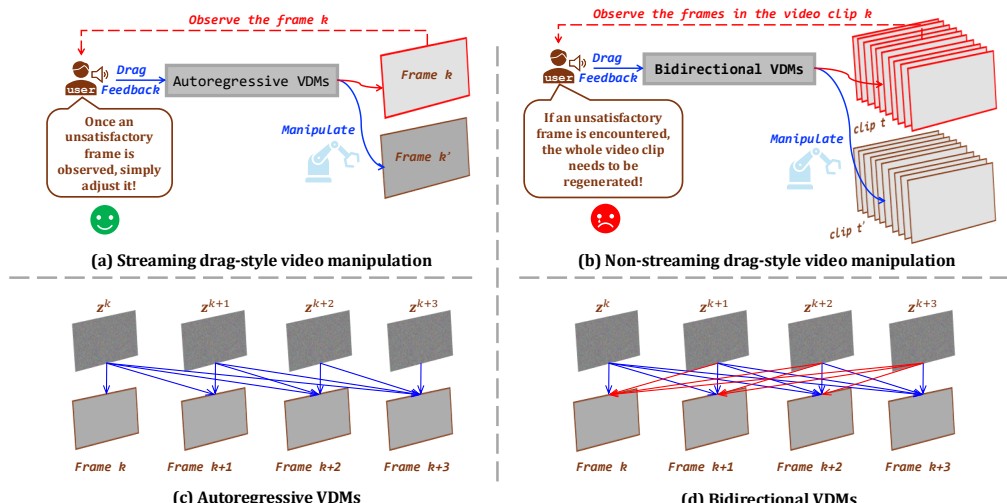

Figure 11: **Comparison between streaming and non-streaming drag-style video manipulation.**

*turn necessitates collecting a large-scale, fine-grained drag-style dataset and finetuning VDMs on it by hundreds or even thousands of H100 GPU hours Yin et al. (2025); Huang et al. (2025).*

In addition to the type of VDMs, another key difference lies in the manipulation process. In streaming drag-style video manipulation, if users find a video frame unsatisfactory and wish to edit it, they only need to apply drag operations to that specific frame. In contrast, in non-streaming drag-style video manipulation, users must provide drag operations for the all subsequent frames to maintain cross-frame consistency, since information flow in bidirectional VDMs is bidirectional as shown in Figure 6 (d), i.e., subsequent video frames can influence preceding video frames. Also, streaming drag-style video manipulation can animate any video frame during the generation process; however, non-streaming drag-style video manipulation struggles to animate intermediate frames, as this conflicts with the bidirectional nature of VDMs, which generates the entire video clip simultaneously. That means animating intermediate frames will break out the consistency of the original video clip.

**REMARK 3.** Tora and DragAnything can be directly integrated with existing autoregressive VDMs as external modules to animate generated video frames. However, they indeed cannot realize streaming control over autoregressive VDMs as they do not alter the original generation direction of VDMs. Differently, our DragStream modifies the latent embeddings of autoregressive VDMs by performing iterative latent region optimization, thereby enabling streaming control over the video generation process by propagating the information of modified latent code through a sliding context window.

**The above analysis demonstrates the importance of advancing existing approaches from non-streaming to streaming drag-style manipulation, highlighting the significant application value of our REVEL task!**

## E    MORE VISUALIZATION RESULTS

In Figure 12, we provide more visualization results achieved by our DragStream approach on our proposed REVEL task. These experimental results still consistently demonstrate that our method can achieve high-quality streaming manipulation over the outputs of VDMs, including both `Editing` and `Animation` with the effects of drag operations such as translation ("`Trans`"), deformation ("`Defor`"), and rotation ("`Rot`"). For more results, please refer to our anonymous project page, which is provided below the abstract.

## F    VISUALIZED ANALYSIS OF SWITCHABLE FREQUENCY

In Figure 13, we investigate the influence of switchable frequency used in SFS during latent region optimization. From the figure, the use of only low-frequency ("$\omega = 0.2$") information easily leads

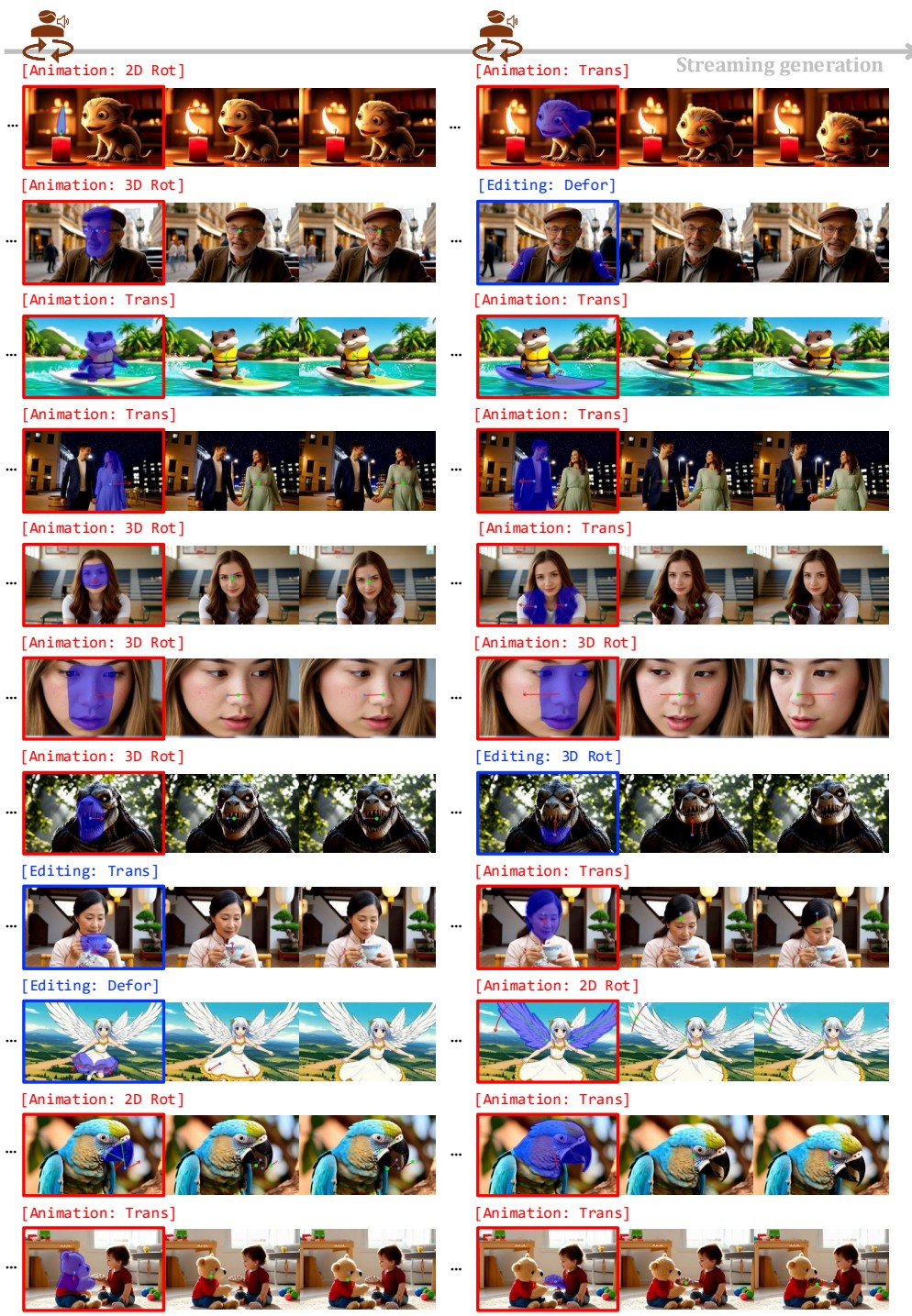

Figure 12: **More visualization results achieved by our DragStream on the REVEL task.** Note that `Editing` produces only one video frame, but we insert an extra subsequent frame to maintain layout consistency with `Animation`.

to scene blurring and unnatural object shape variation. Meanwhile, the high-frequency information inherently contained in original images ("$\omega = 1$") causes noticeable artifacts in generated videos. In contrast, our proposed switchable frequency strategy can balance frequency components, effectively

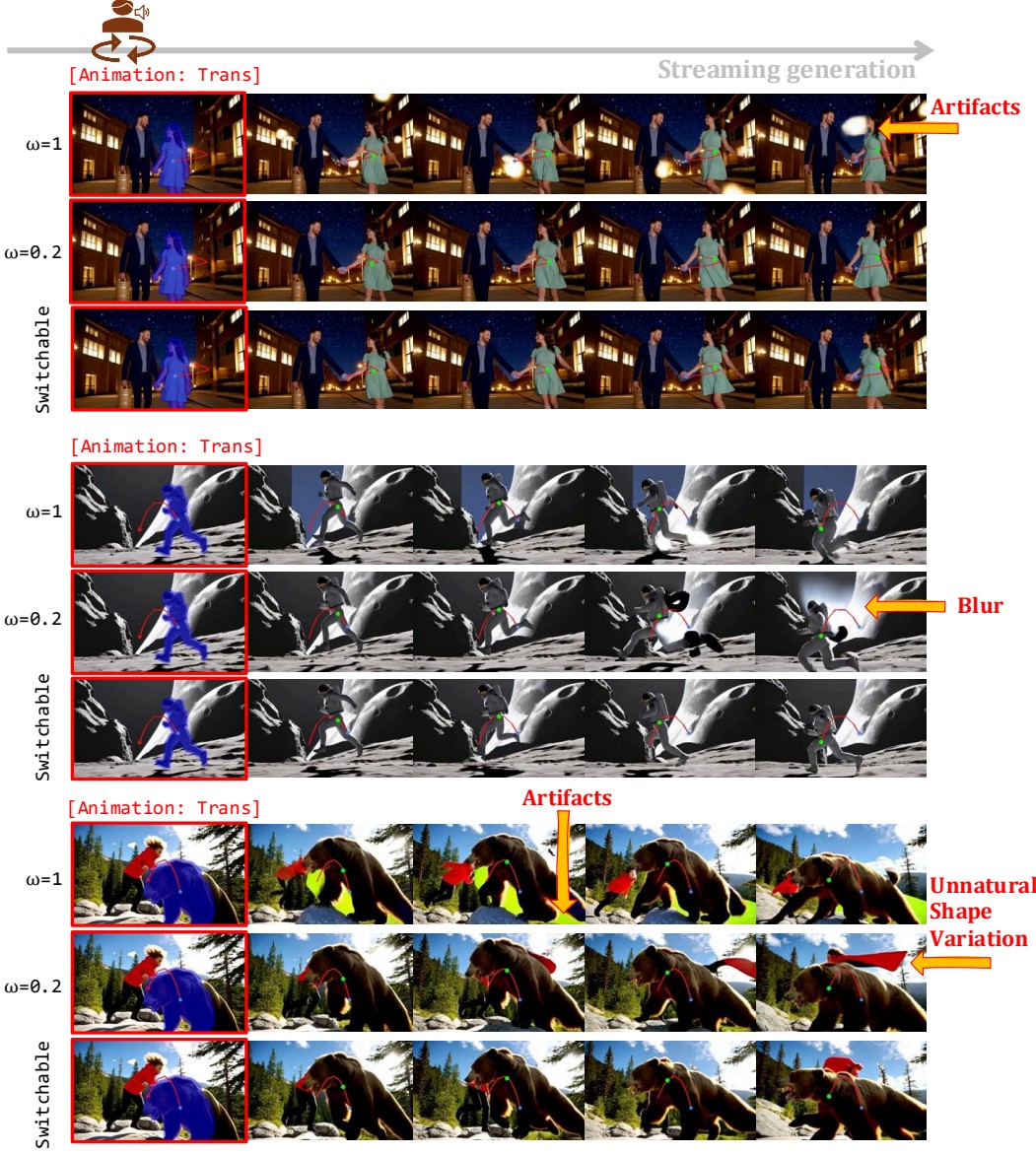

Figure 13: **Visualization analysis of switchable frequency.** $\omega$ **indicates cutoff frequency.**

suppressing artifacts and blur while maintaining object shape consistency. These results confirm that our SFS strategy is crucial for achieving high-quality streaming video manipulation.

# G VISUALIZED ANALYSIS OF GAUSSIAN FILTERING MAP

Figure 14 illustrates the effect of Gaussian filtering maps on streaming drag-style video manipulation. Without the use of Gaussian filtering map ("w/o CSS") to constrain the latent optimization, back-propagated gradient maps may be leaked to irrelevant regions, thereby resulting in artifacts in generated video frames (e.g., distortions around non-target areas). In contrast, with Gaussian filtering maps ("w/ CSS"), gradients are constrained to focus on the most important target region, therefore suppressing their interference to surrounding areas and effectively improving video quality.

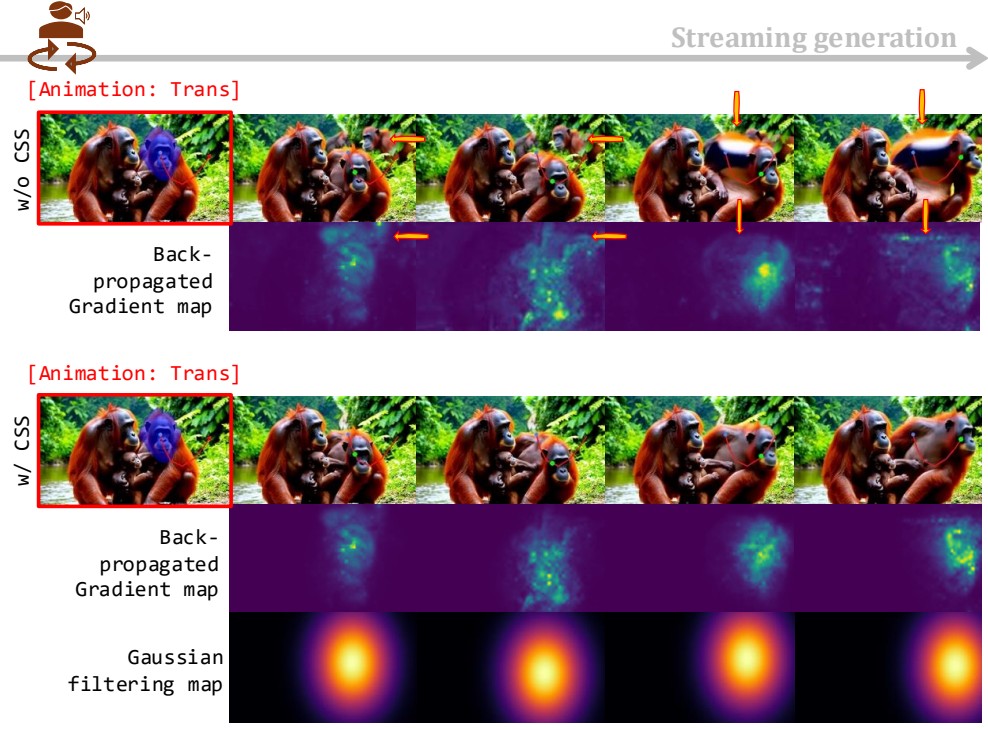

Figure 14: **Visualization analysis of Gaussian filtering map.**

## H    ABLATION STUDY ON OBJECTIVE FUNCTION $\mathcal{L}_{\text{Tot}}$

Table 3: **Ablation study on the objective function used in latent region optimization.**

| Experiment | ObjMC ($\downarrow$) | FVD ($\downarrow$) | FID ($\downarrow$) | DAI ($\downarrow$) |
|---|---|---|---|---|
| w/ $\mathcal{L}_{\text{Rec}} + \mathcal{L}_{\text{Cst}}$ | 26.12 | 596.51 | 25.16 | 0.0545 |
| w/o $\mathcal{L}_{\text{Cst}}$ | 20.87 | 949.06 | 33.55 | 0.0509 |
| w/o $\mathcal{L}_{\text{Rec}}$ | 90.39 | 301.74 | 14.11 | 0.1337 |

In Table 3, we provide ablation studies for the objective function used during latent region optimization. As can be seen from the table, removing the reconstruction loss $\mathcal{L}_{\text{Rec}}$ causes an obvious performance drop in ObjMC and DAI, which indicates that objects in handle regions are not successfully dragged to target points. Although removing the constraint term $\mathcal{L}_{\text{Cst}}$ leads to a slight improvement in ObjMC and DAI, it causes a significant degradation in FVD and FID. This is because, without $\mathcal{L}_{\text{Cst}}$, non-editable regions are severely affected by the latent region optimization, resulting in noticeable artifacts in generated videos. These artifacts significantly degrade the overall video quality, leading to a substantial decline in FVD and FID scores. This observation underscores the importance of $\mathcal{L}_{\text{Cst}}$ in maintaining the integrity of non-editable regions and ensuring high-quality video generation.

## I    COMPATIBILITY OF DRAGSTREAM

In principle, our DragStream method is model-agnostic and can be seamlessly integrated with different autoregressive VDMs. To demonstrate this, we additionally apply our method to the recent autoregressive VDM, CausVid Yin et al. (2025). As can be seen from Figure 15, it can still achieve high-quality streaming drag-style video manipulation, enabling both Editing and animation with fine-grained drag operations such as translation (Trans), deformation ("Defor"), and rotation ("Rot"). These results demonstrate the effectiveness of our proposed method again, and also

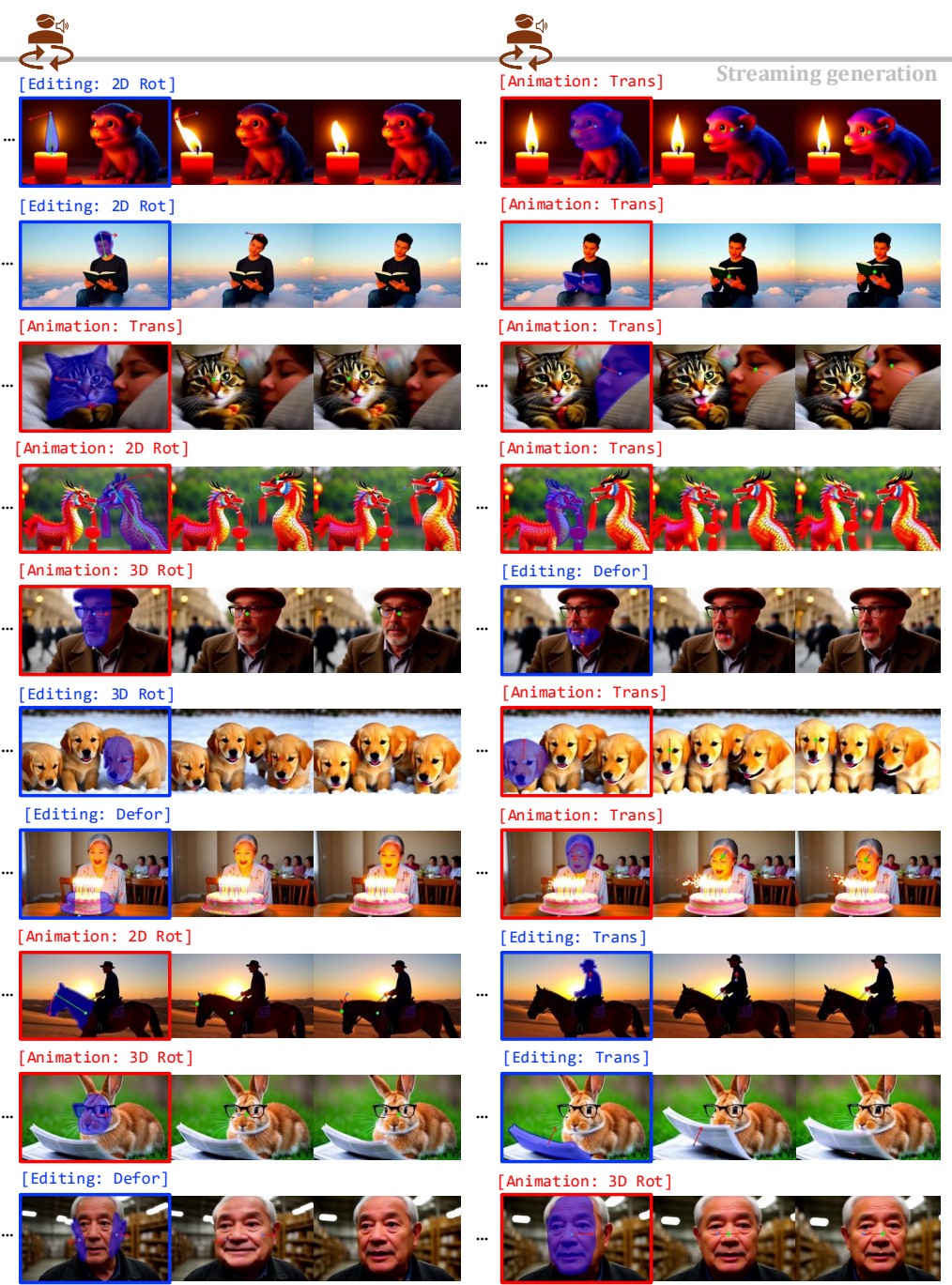

Figure 15: **Visualization results achieved by our DragStream approach based on CausVid Yin et al. (2025).**

highlight its potential as a versatile training-free solution for streaming drag-style video manipulation across different VDM backbones. For more visualization results, we encourage readers to visit our anonymous project webpage, provided below the abstract.

## J    DRAG-STYLE OPERATION TYPES

Currently, our DragStream supports both editing and animating video frames, with user-specified drag effects including translation, deformation, and rotation. For both editing and animation tasks, we can consider the following rules: *i*) translating an object along a trajectory can be achieved by moving its entire region (*as shown in the fourth row from the bottom of Figure 12, the cup translation can be achieved by moving its entire region*); *ii*) deforming object shape can be realized by translating only its edge region (*an example is shown in the third row from the bottom of Figure 12*); *iii*) object 2D rotation can be realized by using a planar rotation transformation (*an example is given in the third row from the bottom of Figure 12*); and *iv*) 3D rotation can be regarded as translating a sub-region of the object, assisted by the inherent prior knowledge of VDMs (*as shown in the second row of Figure 12, the 3D rotation of the man's face can be realized by moving face's sub-region*).

## K    WHY NEED CONTEXT FRAMES?

In section 2, we argue that the context frames are sources of disturbance during streaming drag-style video manipulation. However, we also emphasize that context frames provide crucial visual cues that are essential for subsequent video generation. In Figure 16, we provide a visualization analysis to illustrate the importance of context frames. As shown in the figure, without context frames ("w/o context"), the generated video frames become totally blurry and unnatural. Besides, the foreground object and background scene are completely changed to be inconsistent with the original video. In contrast, with context frames ("w/ context") and our SFSO mechanism, the generated video frames preserves the appearance and structure of the foreground object, as well as the background scene, while achieving high-quality drag-style manipulation. These results firmly demonstrate that context frames provide indispensable visual cues for streaming drag-style video manipulation.

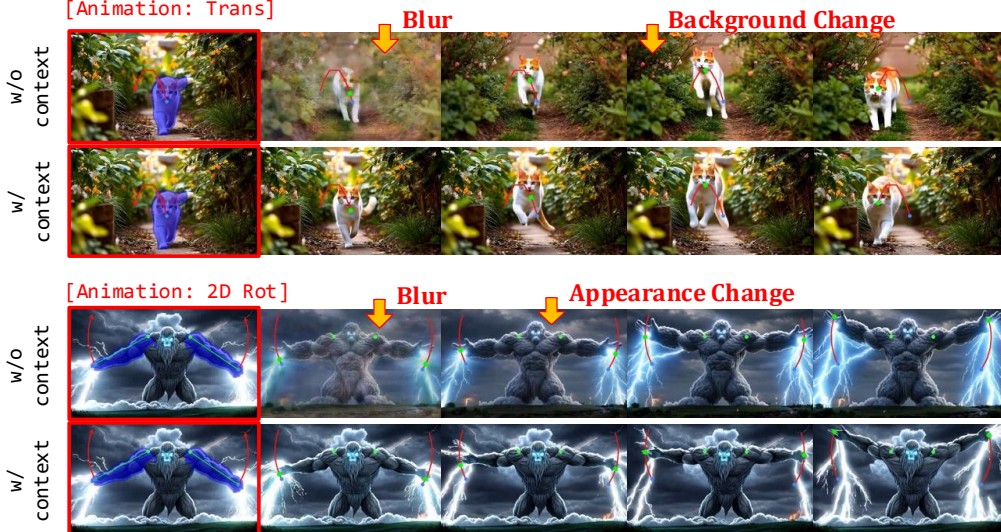

Figure 16: **Visualization analysis on the importance of context frames.**

## L    ADDITIONAL VIDEO RESULTS IN SUPPLEMENTARY MATERIALS

We present additional video results in our supplementary material 'supp-material-1676.zip', including 9 videos, which provide a more intuitive demonstration of the results achieved by our approach. For more visualization results, we recommend readers again to visit our anonymous project webpage: DragStream Demo.

## M    ROBUSTNESS ANALYSIS OF HYPERPARAMETERS

In this section, we analyse the robustness of several hyperparameters in our DragStream.

**Analysis of $L_n$.** The hyperparameter $L_n$ determines how many neighboring latent embeddings are used to rectify drifted latent distribution during performing ADSR. The results in Table 4 shows that our method is robust to the hyperparameter $L_n$. When varying the value of $L_n$ from 3 to 18, our DragStream exhibits consistently stable performance. By contrast, discarding ADSR (i.e., $L_n = 0$) leads to significant performance degradation, which demonstrate the effectivness of our ADSR again.

Table 4: **Analysis of the hyperparameter $L_n$.**

| Experiments | ObjMC ($\downarrow$) | DAI ($\downarrow$) | FVD ($\downarrow$) |
|---|---|---|---|
| $L_n = 0$ | 45.09 | 0.093 | 608.63 |
| $L_n = 3$ | 25.88 | 0.052 | 570.41 |
| $L_n = 9$ | **23.05** | **0.051** | **552.39** |
| $L_n = 18$ | 24.37 | 0.051 | 554.25 |

**Analysis of $\alpha$.** The hyperparameter $\alpha$ determines the shape of the Gaussian filtering map. In Table 5, we provide an analysis of $\alpha$. On one hand, similar to the hyperparameter $L_n$, our DragStream still achieves stable performance when varying the value of $\alpha$ from 2 to 0.5. These results demonstrate that our approach is robust to the hyperparameter $\alpha$. On the other hand, the removal of the Gaussian filtering map ("w/o CSS") results in an obvious performance drop, highlighting the importance of using the Gaussian filtering map to guide the model toward critical areas during drag-oriented manipulation.

Table 5: **Analysis of the hyperparameter $\alpha$.**

| Experiments | ObjMC ($\downarrow$) | DAI ($\downarrow$) | FVD ($\downarrow$) |
|---|---|---|---|
| w/o CSS | 26.12 | 0.061 | 25.16 |
| $\alpha = 2.0$ | 24.70 | 0.054 | 24.52 |
| $\alpha = 1.5$ | 23.50 | 0.052 | 24.17 |
| $\alpha = 1.0$ | 23.65 | 0.053 | 24.04 |
| $\alpha = 0.5$ | 24.52 | 0.055 | 23.65 |

**Analysis of $\omega$.** During latent region optimization, we let the model randomly switch among a set of predefined cutoff frequencies $\omega \in \{0.2, 0.4, 0.6, 1\}$, thereby preventing the high-frequency components from dominating the drag process and introducing artifacts or unnatural results. In Table 6, we show that our method is robust to the set of cutoff frequencies. As can be seen from the table, despite that the low-frequency component is randomly selected from a continuous range $[0, 1, 0.8]$, $[0, 1, 0.7]$, or $[0, 1, 0.6]$, our method still produce stable performance.

Table 6: **Analysis of the hyperparameter $\omega$.**

| Experiments | ObjMC ($\downarrow$) | DAI ($\downarrow$) | FVD ($\downarrow$) |
|---|---|---|---|
| $\omega \in [0.1, 0.8] \cup \{1\}$ | 24.27 | 0.054 | 584.21 |
| $\omega \in [0.1, 0.7] \cup \{1\}$ | 24.05 | 0.054 | 578.65 |
| $\omega \in [0.1, 0.6] \cup \{1\}$ | 23.99 | 0.053 | 558.97 |
| $\omega \in \{0.2, 0.4, 0.6, 1\}$ | 23.05 | 0.051 | 552.39 |

**Analysis of $I$.** Our DragStream is based on an iterative optimization scheme. In Table 7, we investigate the influence of the hyperparameter $I$. The experimental results in the table demonstrate the robustness of our approach to hyperparameters again. For example, despite setting $I$ with a small value, such as $I = 3$ or 2, our approach still exhibits robust performance in streaming drag-style manipulation, which is significantly better than that of the baseline without using our DragStream (i.e., $I = 0$).

Table 7: **Analysis of the hyperparameter $I$.**

| Experiments | FID | ObjMC ($\downarrow$) | DAI ($\downarrow$) |
|---|---|---|---|
| $I = 0$ | **14.11** | 90.39 | 0.133 |
| $I = 2$ | 25.88 | 27.67 | 0.054 |
| $I = 3$ | 24.17 | 24.55 | 0.053 |
| $I = 4$ | 23.72 | **23.05** | **0.051** |

## N    RESULTS ON CONTROLLING BIDIRECTIONAL MODELS

Although our DragStream is designed for autoregressive VDMs, it can also be applied to bidirectional VDMs to control their generation process via drag-style control signal. From the results in Figure 17, we can see that our method can successfully guide the bidirectional model Wan-2.1 to generate videos that conform to drag-style conditions specified by users.

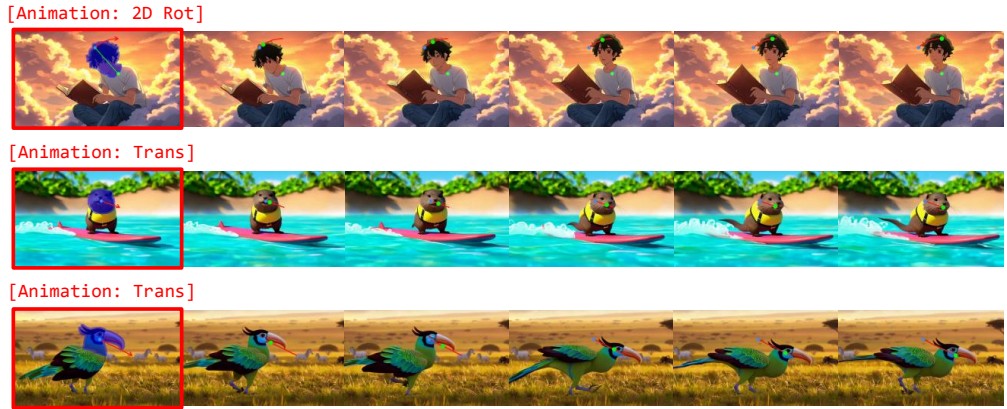

Figure 17: **Results of controlling the video generation of the bidirectional model Wan2.1 via drag-style instructions using our DragStream approach.**

## O   DISCUSSION OF PROMPT–DRAG CONFLICTS

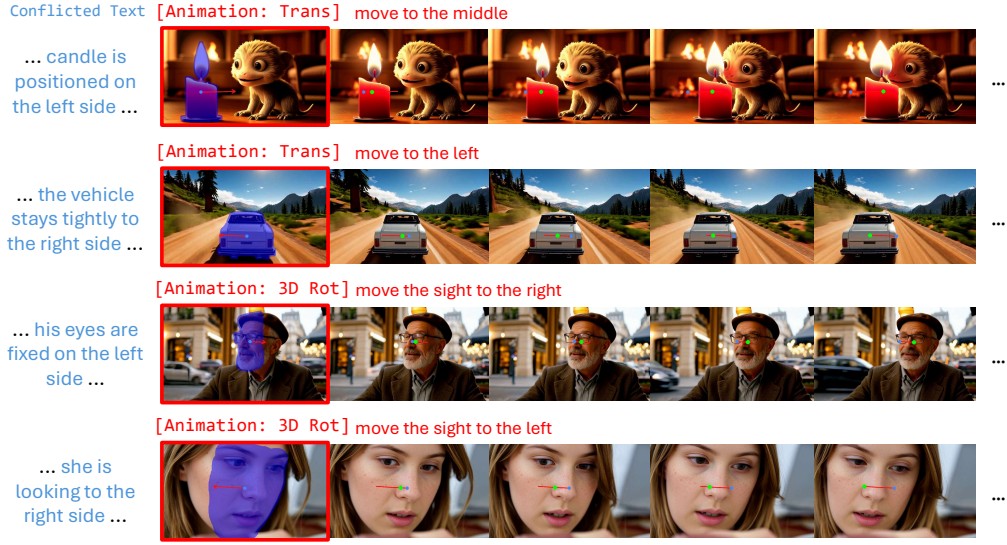

Figure 18: **Visualized case study on prompt-drag conflicts.**

In Figure 18, we further investigate an interesting scenario in streaming drag-style video manipulation where drag operations conflict with text prompts. We observe that when such conflicts occur, VDMs consistently follow drag instructions specified by users during generation. This is because our DragStream revises latent embeddings more explicitly. For example, as shown in the figure, although the text prompt requires the car to remain on the right side, our DragStream can successfully drag the car toward the left side of the path.

## P   ADDITIONAL RESULTS ON LONG VIDEO GENERATION

In this section, we provide additional streaming drag results of our DragStream in long video generation. As shown in Figure 19, despite accumulated errors remain a challenging issue for current autoregressive VDMs, our method can still effectively realize drag-based manipulation in 5s, 10s, and 20s. Importantly, compared with the original videos generated by Self-Forcing Huang et al. (2025), drag operations introduced by our DragStream do not degrade video quality either during the drag-based manipulation process or in subsequently generated video frames after the manipu-

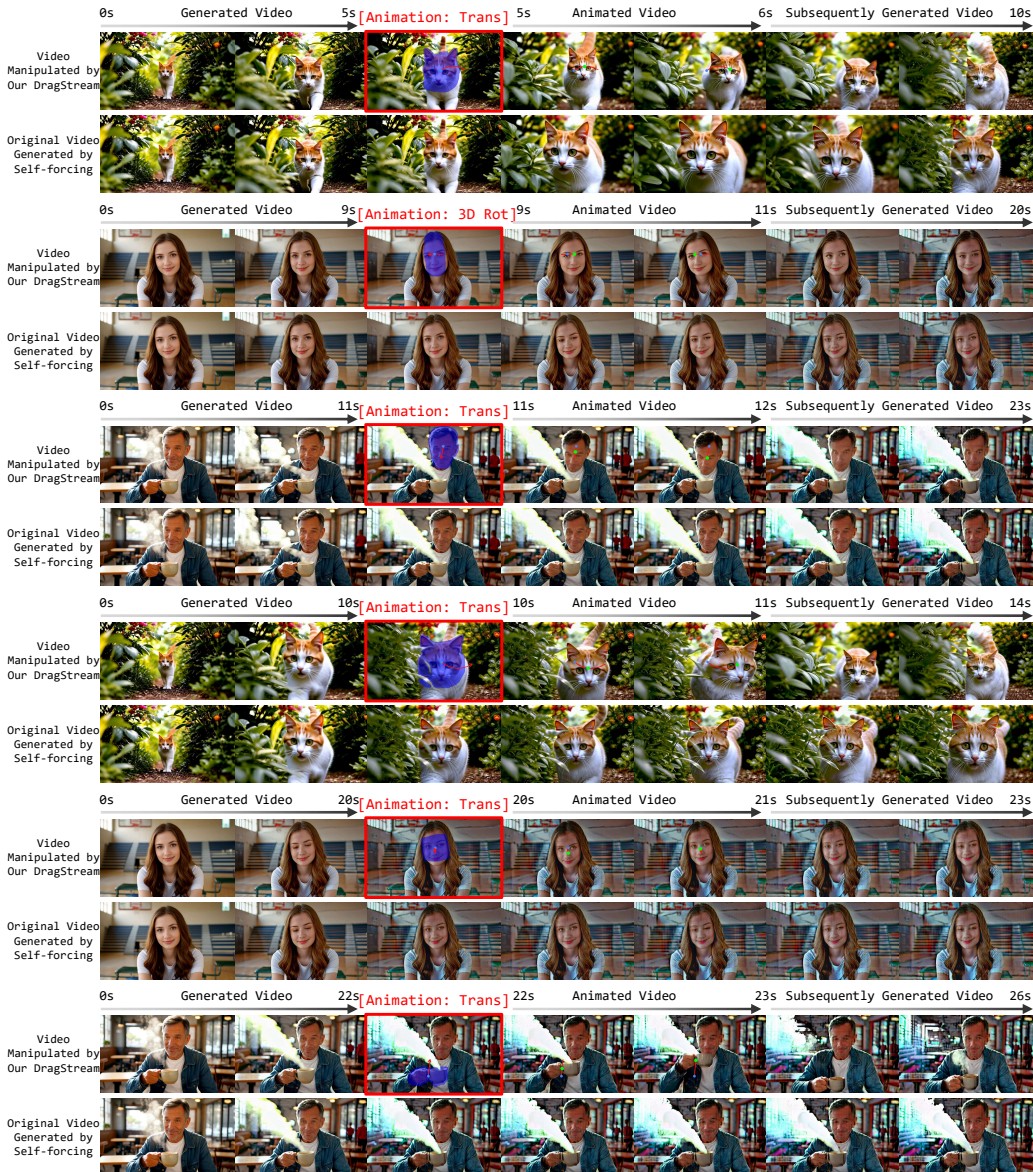

Figure 19: **Additional results of our DragStream on long video generation.**

lation. For example, in the second case, when we manipulate the frame at 9s (i.e., dragging the woman's face to the left), the animated video clip maintains the same quality as that produced by Self-Forcing. Moreover, after the manipulation, our DragStream can continue to preserve the same quality as Self-Forcing in the subsequently generated frames from 11s to 20s. These experimental results firmly demonstrate that the effectiveness of our DragStream approach in streaming long-video generation.

## Q  LONG-DURATION DRAG-ORIENTED MANIPULATION

In this section, we further investigate the effectiveness of our ADSR strategy in long-duration drag-oriented video manipulation. As shown in Figure 20, performing drag-style manipulation without ADSR ("w/o ADSR + Drag") results in severe latent distribution drift, which leads to noticeable degradation in video quality compared with the baseline without drag-based manipulation ("Ref + w/o Drag"). In contrast, our ADSR ("w/ ADSR + Drag") can effectively suppresses the latent

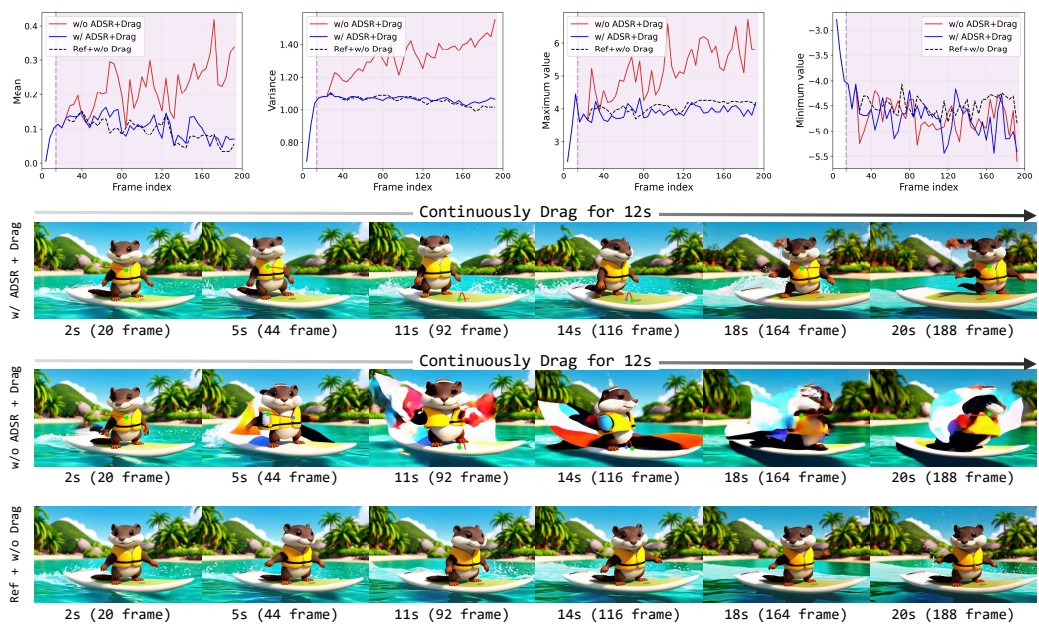

Figure 20: **Analysis of ADSR in long-duration drag-oriented manipulation.**

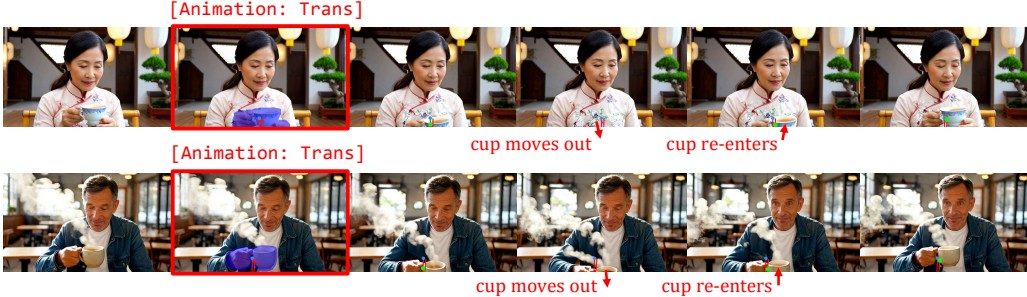

Figure 21: **Streaming drag with object leaving and re-entering.**

distribution drift issue even when drag operations are sustained for over 20s. Consequently, the video generated by our DragStream ("w/ ADSR + Drag") is significantly better than that of "w/o ADSR + Drag", further demonstrating the effectiveness of our DragStream approach.

## R  OBJECT LEAVING AND RE-ENTERING

In addition to object occlusion and re-emergence discussed in Section 5.3, we also evaluate our DragStream in another interesting scenario—objects leaving and re-entering the view. As shown in Figure 21, when an object is dragged to move out of the frame and later back into view, DragStream effectively preserves its appearance and structure, producing high-quality results of drag-based manipulation. Actually, the re-entering process is essentially no different from the standard drag-based manipulation. We just need to save the latent features of an object before it moves out of view, and then reconstruct is along a user-given trajectory via latent optimization.

