# OpenReview forum: "Streaming Drag-Oriented Interactive Video Manipulation: Drag Anything, Anytime!"
_ICLR.cc/2026/Conference — ICLR 2026 Poster_

### Official Review · Reviewer_y2YG · 2025-10-16

**Soundness:** 2
**Presentation:** 3
**Contribution:** 2
**Rating:** 2
**Confidence:** 4

**Summary:**

This paper presents the REVEL task, designed to enable users to edit the future generation content by conducting drag-style editing on the existing last frame. It unifies the editing and animation tasks under one framework, allowing for user-specified translation, deformation, and rotation effects on general objects during generation. This paper proposes the DragStream framework to solve the REVEL task. DragStream contains ADSR to handle distribution shift and SFSO to mitigate context inference from previous frames.

**Strengths:**

The strengths of the framework can be summarized as follows:

1. Interesting task proposed: The REVEL task that allows the user to modify anything on anything that unifies drag-editing and animation.
2. Both ADSR and SFSO are well-structured and decently presented methods
3. The experiment and the appendix section provide a relatively comprehensive view of the performance of the DragStream framework

**Weaknesses:**

Despite the paper presenting an interesting task and relatively comprehensive experiment results. However, the weakness of this paper can not be ignored. The weaknesses are as follows:

1. **Unclear and impractical task definition:** Despite being conceptually novel, I think the REVEL task is not clearly or realistically defined. The task claims to have the effect of modifying the generated videos at any time and for anything. This creates an internal contradiction in the task definition and leads to practical limitations. In particular, editing only future frames can cause severe temporal discontinuities between past and modified content, violating one of the central goals of video generation—temporal consistency. This is my major concern to the paper.
2. **Overclaiming effect:** The paper claims that the REVEL task allows users to edit anything at any time. However, the DragStream framework can not support that. It only supports three ways of editing (namely translation, deformation, and rotation), but is unable to perform broader semantic or structural edits. Moreover, it operates only during the generation process, not on completed videos.
3. **Lack of failure case studies: T**here are no studies of how, why, and under what circumstances the framework would fail. This makes it hard to understand the effect of the framework comprehensively.
4. **No latency analysis:** Since the proposed framework requires the user to edit during video generation, the computation time cost is vital. However, the paper does not report latency, runtime, or GPU cost for editing a single video, which are essential for assessing the practicality of interactive systems.
5. **Complexity in notation:** Although the methodology is detailed, the notation used in the paper is unnecessarily complicated, which hampers readability and accessibility for a broader audience.

**Questions:**

I have the following questions for the authors of the paper to refine their paper further:

1. I'd like to know if there are any methods the authors can propose to propagate editing into previous frames, not just the frames to be generated.
2. Since the paper only provides three ways of editing, I wonder whether it can support general editing instructions.
3. I wonder how, why, and under what conditions the DragStream framework would fail.
4. What is the average time to process a video?

---

> ### Author Response · Authors · 2025-11-21
>
> *Dear Reviewer y2YG,*
>
> &nbsp;
>
> First of all, we sincerely appreciate your valuable questions, suggestions, and comments. Please give me an opportunity to clarify some of your **`misunderstandings`**, which seems to be your major concerns.
>
> &nbsp;
>
> *Best regards,*
> *Authors of paper #1676*

---

> ### Author Response · Authors · 2025-11-21
> **Response to Reviewer y2YG [1/2]**
>
> ### **Q1. Misunderstanding-1: editing only future video frames.**
> **A1.** Thanks for your comments. We must point out that there is a misunderstanding for our paper.
>
> We would like to clarify that **our DragStream does not aim at editing future frames; it indeed aims to `edit or animate existing frames`.**
>
> Our approach considers both editing and animation.
> >1. For editing, users **`first observe the outputs`** of autoregressive VDMs; then, once an  unsatisfactory frame is produced, users can **`immediately`** modify it (streaming but **not future**).
> 2.For animation, users **`first observe a currently generated video frame`**; then, drag instructions given by users are applied to objects in the frame (e.g., turning left/right) to animate and produce a video clip.
>
> **"Drag anything, anytime"** means that once an unsatisfactory frame is produced, users can **`immediately`** modify it via dragging, and accordingly adjust the subsequent generation process, **without having to wait for the end of generating the entire video clip as is required in non-streaming scenarios**.
>
> To some extent, one main advantage of streaming drag-oriented video manipulation is ensuring temporal consistency. **That is because once an unsatisfactory frame is produced, users can `immediately` modify it, and accordingly `alter` the subsequent generation direction by using modified frames as contexts through KV cache.** However, non-streaming models generate **`the whole video simultaneously at once`**. For editing performed after the video has already been fully generated, temporal inconsistency always exist between edited and unedited frames; otherwise, users need to edit all of the frames.
>
> **Finally, we also want to highlight that approach can edit frames that already been completely generated.**  The discussion for this point is provided in the last paragraph of the response to your **Q2**.  Thus, in our approach, there is no temporal inconsistency between an edited frame and its subsequent frames, nor between the edited frame and its preceding frames.
>
> &nbsp;
>
> ---
>
> ### **Q2. Misunderstanding-2: `edit` anything, anytime; `only applicable` during generation.**
> **A2.** Thanks for your comments! We must point out there are some misunderstandings.
>
> First, in our paper, we indeed do not state that our DragStream can **`Edit` Anything, Anytime**; instead, we claim that DragStream can **`Drag` Anything, Anytime**! There is a significant conceptual difference between **"`Edit`"** and **"`Drag`''**. Drag, a low-level control signal, aims to manipulate objects' shape and position, which does not aim at offering high-level control signal, such as semantic or style change. **We must claify that using drag signal to achieve semantic or style editing is `highly uncommon`**. To our best knowledge, there are no prior drag-based methods trying to do this.
>
> Second, we also want to highlight again that **our approach does not aim at editing future frames**; instead, it aims to edit or animate existing frames. So, **the method is absolutely applicable to frames that have already been completely generated.** **`For clearly displaying videos, the results are provided on Section VII of our anonymous` [webpage](https://anonymous-30f12b51.github.io/Streaming-Drag-Oriented-Interactive-Video-Manipulation.github.io/ ).** As shown in Section VII of the [webpage](https://anonymous-30f12b51.github.io/Streaming-Drag-Oriented-Interactive-Video-Manipulation.github.io/ ), when the 60-frame videos are completely generated, we can still manipulate all the previous frames, e.g., dragging the head of the bird upward in all the previous frames. The results of our DragStream on the bidirectional model (Section II of our [webpage](https://anonymous-30f12b51.github.io/Streaming-Drag-Oriented-Interactive-Video-Manipulation.github.io/ )) also demonstrate that our method can be used to modify frames that have already been generated, since the model is not streaming and drag operations are all applied to frames that have been completely generated.

---

> ### Author Response · Authors · 2025-11-21
> **Response to Reviewer y2YG [2/2]**
>
> ### **Q3. Lack of failure case studies.**
> **A3.** Thanks for your questions. Some failure cases are added in the manuscript. **We want to highlight this is an indeed minor issue.** Actually, our method cannot produce highly unreasonable or physically implausible videos. This is because the underlying VDMs are trained on large-scale real-world video datasets. When drag instructions are highly unreasonable or physically implausible, our approach will fail produce high-quality manipulation results. For better displaying videos, we show the results on our anonymous [webpage](https://anonymous-30f12b51.github.io/Streaming-Drag-Oriented-Interactive-Video-Manipulation.github.io/ ) (Section VIII).
>
> &nbsp;
>
> ---
>
> ### **Q4. Runetime analysis.**
> **A4.** Thanks for your good question! In fact, DragStream is **sufficiently fast while also being friendly in computational resources (`only requiring one GPU card`)**. In the table below, we analyze the runtime under the different iteration numbers of the latent optimization ($I=4$, $3$, and $2$). Actually, our method indeed only brings acceptable runtime overhead compared with the baseline ("w/o our method"). All our experiments are conducted on only **`a single`** H20 GPU card, and runtime, ObjMC, and DAI are averaged over 204 video clips.
>
> &nbsp;
>
> | Experiments                 | Runtime/frame ↓ | ObjMC   ↓| DAI  ↓ |
> |:----------------------------:|:---------------:|:-----------:|:--------:|
> | $I = 4$                    | 0.30s          | 23.05     | 0.051  |
> | $I = 3$                    | 0.27s          | 24.55     | 0.053  |
> | $I = 2$                    | 0.24s          | 27.67     | 0.054  |
> | $I = 0$ (w/o our method) | 0.17s          | 90.39     | 0.133  |
>
> &nbsp;
>
> The above results domenstrate that the efficiency of our approach is indeed acceptable. **Kindly note that H20 is an outdated GPU type; the use of A100 or H200 can even achieve faster speed and higher efficiency.**
>
> &nbsp;
>
> ---
>
> **Q5. Complexity in notation.**
> **A5.**  We sincerely appreciate for pointing out this issue and also feel sorry about that. As stated in Proposition 1 (line 143~156), **this paper aims to offer a unified perspective for current drag-style video manipulation tasks and establish a new standard for subsequent works.** So, it try to enable users to perform both **`editing`** and **`animation`** on video frames via drag-style operations, each supporting user-specified **`translation`**, **`deformation`**, and **`rotation`** effects. Actually, it is not easy to use very few notations to clearly illustrate the whole process, especially some are needed to be presented in formulas. **Since our work is the first attempt trying to do this, we believe a rigorous symbol system is necessray**. So, we clearly explain the main notations in Table 1 of Section B. We find that solely reducing notations are somewhat inappropriate. Oversimplified notations may introduce ambiguity, since notations can be simply removed, but the underlying concepts, operations, and our strategies still exist and cannot be removed.
>
> **`Thanks again for your valuable comments. We promise we will try our best to polish our manuscript!`**

---

> > ### Comment · Reviewer_y2YG · 2025-11-26
> > **Thanks for your rebuttal**
> >
> > Dear Authors,
> >
> > Thank you so much for the clarification; it makes the paper much clearer. I also appreciate your efforts in creating a webpage to demonstrate your idea. Although I am also somewhat skeptical about editing the intermediate frames, I still increased the score.
> >
> > Best,
> > Reviewer ywYG

---

> ### Author Response · Authors · 2025-11-27
> **Reply to Reviewer y2YG**
>
> *Dear Reviewer y2YG,*
>
> &nbsp;
>
> **We sincerely appreciate your decision to raise the score!**
>
> **If possible, would you be willing to elaborate on your specific concerns?**
> We are **`confident and very willing`** to address all of your questions and concerns.
>
> Actually, we would like to highlight again that there is **`no essential difference`** between editing currently generated frames and editing previously generated frames—both only require modifying their corresponding latent embeddings.
>
> **Once again, we sincerely thank you for your valuable suggestions, comments, and questions, as well as for your decision to raise the score---*which truly means a great deal to us*!**
>
>
> &nbsp;
>
> *Best regards,*
> *Authors of paper #1676*

---

> ### Author Response · Authors · 2025-11-28
> **Looking forward to receiving further feedback**
>
> *Dear Reviewer y2YG,*
>
> &nbsp;
>
> Thanks again for your comments that truly mean a great deal to us. **`Would you be willing to elaborate on your specific concerns?`** We would like to remind again that **we are very confident and willing to address all of your questions and concerns**.
>
> There indeed is no essential difference between editing **the currently generated frame** and editing **previously generated frames**—**both are existing frames**. Although our DragStream focuses on streaming manipulation, it can absolutely edit previously generated frames—**either by manually providing user-specified drag instructions (the top of Section VII) to these previous frames or by using the tracking strategy employed in **`DragVideo` to propagate the current frame's drag trajectory to previous frames (the bottom of Section VII)**.** *Again, we would like to emphasize that the requirement for editing both currently generated and previously generated frames is the same—revising the latent embeddings of video frames according to drag operations given by users.*
>
> **If there are any remaining issues or if further clarification is needed on any point, we are more than willing to address them immediately.**
>
> &nbsp;
>
> *Best regards,*
> *Authors of paper #1676*

---

### Official Review · Reviewer_gQWk · 2025-10-30

**Soundness:** 3
**Presentation:** 3
**Contribution:** 3
**Rating:** 6
**Confidence:** 4

**Summary:**

This paper introduces a new task, streaming drag-oriented interactive video manipulation and a corresponding training-free framework called DragStream. The goal is to enable fine-grained, real-time video control via drag-style user interactions, allowing users to modify generated videos anytime on any object. To address two key challenges—latent distribution drift and context interference—the paper proposes two techniques: Adaptive Distribution Self-Rectification (ADSR), which stabilizes latent features using neighboring frame statistics, and Spatial-Frequency Selective Optimization (SFSO), which mitigates contextual noise through selective propagation in spatial and frequency domains.

**Strengths:**

* The work defines a clear and meaningful new task that formalizes streaming, interactive drag-style manipulation, a setting not explicitly covered by prior works.
* The proposed DragStream method is conceptually elegant and computationally efficient, requiring no finetuning while offering plug-and-play integration with autoregressive diffusion models.
* The paper is well-written, and the figures effectively convey both the problem and the proposed solution.

**Weaknesses:**

* Evaluation is limited to adapted baselines that are not optimized for streaming settings, which makes it difficult to judge true generalization advantages.
* The framework depends heavily on empirical hyperparameters, e.g., cutoff frequencies, Gaussian filtering spread, without deeper theoretical justification or convergence analysis. And these empirical hyperparameters may harm its generalization ability.
* The paper focuses on short clips and relatively simple object manipulations. However, in real streaming setups, drag edits often accumulate over long sequences. The manuscript does not examine whether distribution drift resurfaces over extended temporal spans or whether contextual filtering introduces temporal lag or smoothing artifacts.
* Although the method is claimed to be “training-free,” the iterative latent-region optimization may still incur noticeable inference overhead. The paper lacks detailed runtime analysis, such as per-frame latency under streaming conditions, GPU memory usage, or responsiveness to real-time drag input.

**Questions:**

Most of my concerns have been discussed above, while I still have several minor questions as follows,

* Can the authors provide more insight into how ADSR interacts with the diffusion denoising schedule? For example, does applying rectification at different timesteps change the generation stability?

* How does the framework handle object occlusions or re-appearance during dragging? Is there any mechanism to prevent artifacts when an object moves out of view and re-enters the frame?

---

> ### Author Response · Authors · 2025-11-21
> **Response to Reviewer gQWk [1/3]**
>
> ### **Q1. Baselines are not originaly designed for the streaming setting.**
> **A1.** Thank you for your comments! We would like to emphasize that **our paper is `the first work` to address streaming drag-oriented interactive video manipulation from a `unified` perspective** (supporting both frame **`editing`** and **`animation`** with user-specified **`translation`**, **`deformation`**, and **`rotation`** effects). As a results, no appropriate approach has been specifically designed to tackle this problem. **We believe that your comments also reflect the `value` of our work, and `the urgency` of paving the way to unified streaming drag-oriented video manipulation!**
>
> We also want to kindly remind that our DragStream approach is successfully applied to both **`Self-Forcing`** and **` CausVid`**. In fact, **Self-Forcing and CausVid are really recent typical streaming VDMs**, and are significantly different from the paradigm of conventional bidirectional VDMs by **`generating videos in a streaming way`**. The high-quality results on these models can solidly demonstrate our method's effectiveness in realizing drag-oriented manipulation in a streaming manner.
>
> &nbsp;
>
> ----
>
> ### **Q2. Cutoff frequencies and Gaussian filtering spread.**
> **A2.** Thanks for your valuable comments!
>
> We would like to point out that **hyperparameters are `unavoidable`** in modern deep learning techniques and models. But we want to clarify that **our approach is indeed `robust` to the main hyperparameters**, such as cutoff frequencies and Gaussian filtering spread. The experimental results are shown below.
>
> #### **1. Cutoff frequencies**
> In our paper, **we introduce three additional low frequencies starting from 0.2 and `increasing at a fixed interval` of 0.2**, namely 0.2, 0.4, and 0.6. We do not carefully select or tune these values. To demonstrate the robustness of our method to these hyperparameters, **we turn to randomly selecting them from `the continuous range [0.1, 0.8]` during latent region optimization in each iteration**. According to the results in the following table, we can see that our method still achieves stable performance. This also aligns with the **`philosophy`** of our approach: **we do not aim to capture any specific frequencies; instead, we seek to prevent high-frequency components from dominating the drag process and to avoid the model overfitting to noisy patterns in high frequency.**
>
> &nbsp;
>
> | Cutoff frequencies | ObjMC ↓ | DAI  ↓ | FVD ↓|
> |:------------------:|:-----:|:------:|:------:|
> | 0.2, 0.4, 0.6, 1    | 23.05 | 0.051 | 552.39 |
> | [0.1, 0.8], 1       | 24.27 | 0.054  | 584.21 |
>
> &nbsp;
>
> #### **2. Gaussian filtering spread**
> In the following table, we also conduct experiments for the hyperparameter $\alpha$ that controls the spread of the Gaussian filtering map. **The results demonstrate the robustness of our approach again!**
>
> &nbsp;
>
> | $\alpha$          | ObjMC ↓ |   DAI  ↓  |  FVD ↓ |
> |:-----------------:|:---------:|:--------:|:-------:|
> | 2                 | 24.70   | 0.054   | 24.52   |
> | 1.5               | 23.50   | 0.052   | 24.17   |
> | 1.0 (ours)        | 23.65   | 0.053   | 24.04   |
> | 0.5               | 24.52   | 0.055   | 23.65   |
> | w/o Gaussian filtering map  | 26.12 | 0.061   | 25.16   |
>
>
> &nbsp;
>
> ----
>
> ### **Q3. Edit operations accumulate over long sequences.**
> **A3.** Thanks for your valuable comments! Thanks to the use of ADSR, our approach can still effectively suppress distribution drift, even when drag operations accumulate over long sequences.  **`For clearly displaying videos, the results are provided on Section IV of our anonymous` [webpage](https://anonymous-30f12b51.github.io/Streaming-Drag-Oriented-Interactive-Video-Manipulation.github.io/)** As shown in Section IV of our [webpage](https://anonymous-30f12b51.github.io/Streaming-Drag-Oriented-Interactive-Video-Manipulation.github.io/), even under the drag operation lasting about 20 seconds, our method can still produce high-quality results, with the target object timely following our drag instructions. Meanwhile, the distribution drift issue remains effectively suppressed. These results demonstrate the effectivness of our method again!

---

> ### Author Response · Authors · 2025-11-21
> **Response to Reviewer gQWk [2/3]**
>
> ### **Q4. Runtime analysis.**
> **A4.**  Thanks for your construcive comments! In fact, DragStream is **sufficiently fast while also being friendly in computational resources (`only requiring one GPU card`)**. In the table below, we analyze the runtime under the different iteration numbers of latent optimization ($I=4$, $3$, and $2$). Actually, our method indeed only brings acceptable runtime overhead compared with the baseline ("w/o our method"). All the our experiments are conducted on only **`a single`** H20 GPU card (our approach consumes only about 40 GiB memory), and runtime, ObjMC, and DAI are averaged over 204 video clips.
>
> &nbsp;
>
> | Experiments                 | Runtime/frame ↓ | ObjMC   ↓| DAI  ↓ |
> |:----------------------------:|:---------------:|:-----------:|:--------:|
> | $I = 4$                    | 0.30s          | 23.05     | 0.051  |
> | $I = 3$                    | 0.27s          | 24.55     | 0.053  |
> | $I = 2$                    | 0.24s          | 27.67     | 0.054  |
> | $I = 0$ (w/o our method) | 0.17s          | 90.39     | 0.133  |
>
> &nbsp;
>
> The above results domenstrate that the efficiency of our approach is indeed acceptable. **Kindly note that H20 is an outdated GPU type; the use of A100 or H200 can even achieve faster speed and higher efficiency.**

---

> ### Author Response · Authors · 2025-11-21
> **Response to minor questions of Reviewer gQWk [3/3]**
>
> ###  **Q5. How does ADSR interact with diffusion denoising timesteps?**
> **A5.** Thanks for your constructive questation! We would like to clarify that latent region optimization is only conducted in **`a single denoising timestep`** ($T'$) with mutiple iterations, and accordingly ADSR is also only performed in the timestep $T'$.
> Performing ADSR on a different timestep other than $T'$ will fail to timely rectify the drag-induced distribution drift of latent embeddings, thus lowering the generation stability of models. **In fact, ADSR can effectively `suppress the distribution drift` of latent embeddings, so as to make the generation process `more stable`, including the improvement of all the `ObjMC`, `DAI`, and `FVD`.** Without the use of ADSR, the generation process is easily affected by the distribution drift issue caused by accumulated perturbations (illustrated in Figure 2 (a) of the main paper).
>
> &nbsp;
>
> | Experiments | ObjMC ↓ |   DAI ↓    |   FVD ↓   |
> |:-----------:|:--------:|:--------:|:---------:|
> | w/o ADSR    | 45.09    | 0.093   | 608.63    |
> | w/ ADSR    | 23.05    | 0.051   | 552.39    |
>
> &nbsp;
>
> ----
>
> ### **Q6. Questions about object occlusion or re-appearance.**
> **A6.** We would like to clarify that **the base autoregressive VDMs are trained on massive amounts of data and therefore learn `rich prior knowledge` about object occlusion.** As a result, when drag instructions involve simple cases of object occlusion, we find that VDMs can inherently handle them effectively. **`To clearly display the videos, results are provided in Section V of our anonymous` [webpage](https://anonymous-30f12b51.github.io/Streaming-Drag-Oriented-Interactive-Video-Manipulation.github.io/)** As can be seen from the results, when dragging down the woman's head, the model can well handle object occlusion between the woman and the cup.
>
> For re-appearance, we just need to **save the latent features of an object** before it moves out of view; then, we can gradually reconstruct the object and let it re-enter the screen. We show the results on Section VI of our [webpage](https://anonymous-30f12b51.github.io/Streaming-Drag-Oriented-Interactive-Video-Manipulation.github.io/). Actually, the re-appearance process is essentially no different from the standard drag-based manipulation (reconstructing objects along a user-given trajectory via latent optimization).

---

> ### Author Response · Authors · 2025-11-28
> **Looking forward to receiving further feedback**
>
> *Dear Reviewer gQWk,*
>
> &nbsp;
>
>
> **We sincerely thank you again for your constructive comments—they truly mean a great deal to us!** We have carefully addressed each of your comments with detailed responses, analyses, and additional experiments. We have also uploaded a revised version of our paper, with all modifications clearly highlighted.
>
> Your insights have been invaluable in helping us strengthen our work! **We would greatly appreciate it if you could let us know whether there are any remaining issues or points that may require further clarification. We would be more than happy to provide any additional explanation you may need!**
>
> Thank you again for your time and valuable feedback. We look forward to hearing from you!
>
> &nbsp;
>
>
> *Best regards,*.
> *Authors of paper #1676*

---

### Official Review · Reviewer_HUjr · 2025-10-30

**Soundness:** 3
**Presentation:** 1
**Contribution:** 3
**Rating:** 6
**Confidence:** 3

**Summary:**

This paper introduces REVEL (stReaming drag-oriented interactive vidEo manipuLation), a new task for fine-grained, interactive control of streaming video generation. The authors propose DragStream, a novel training-free method that enables users to drag any part at any time by manipulating objects within autoregressive video diffusion models. DragStream addresses two key challenges: it uses an Adaptive Distribution Self-Rectification (ADSR) strategy to prevent the accumulating "latent distribution drift" caused by user edits, and it employs a Spatial-Frequency Selective Optimization (SFSO) mechanism to mitigate "context interference" from previous frames. This allows users to perform complex translation, deformation, and rotation operations on streaming video content.

**Strengths:**

The method's primary strengths lie in its practical application and thorough empirical validation.

- The ability to perform streaming, drag-oriented control is a highly useful and interesting application for interactive video generation, moving beyond static, one-off edits.

- The visual results, as demonstrated in the supplementary videos, are impressive and clearly show the method's effectiveness in manipulating video content smoothly and coherently across frames.

- The paper is supported by a strong set of quantitative experiments and a detailed ablation study (e.g., Figures 6 & 7) that clearly validates the contribution and necessity of each proposed component (ADSR and SFSO).

**Weaknesses:**

- I think the main major weakness is the presentation of the paper. The core concepts, which appear to be intuitive, are obscured by an extremely dense and overly complicated notational system. This heavy use on complex notations makes it very difficult for the reader to build intuition and grasp the main contributions without significant, unnecessary effort.

- In the Spatial-Frequency Selective Optimization (SFSO) mechanism, the Switchable Frequency-domain Selection (SFS) strategy (Prop 3) is key. The paper states the cutoff frequency $\omega$ is randomly selected from {0.2, 0.4, 0.6, 1} (an ablation study is also done on fgure 7). Could the authors elaborate on the intuition and provide more explanation?

- For the Adaptive Distribution Self-Rectification (ADSR) strategy (Prop 2), the latent code is rectified using the statistics (${\mu}_{T'}$, ${\sigma}_{T'}$) from preceding neighboring frames. How is the window size for these neighboring frames determined, and how sensitive is the model to this hyperparameter? Is there a trade-off between a larger window (more stable statistics but potentially less relevant) and a smaller one (more relevant but noisier statistics)?

**Questions:**

Please see weaknesses.

---

> ### Author Response · Authors · 2025-11-21
> **Response to Reviewer HUjr**
>
> ### **Q1. Paper's writing.**
> **A1.** We sincerely appreciate for pointing out this issue and also feel sorry about that. As stated in Proposition 1 (line 143~156), **this paper aims to offer a unified perspective for current drag-style video manipulation tasks and establish a new standard for subsequent works.** So, it tries to enable users to perform both **`editing`** and **`animation`** on video frames via drag-style operations, each supporting user-specified **`translation`**, **`deformation`**, and **`rotation`** effects. Actually, it is not easy to use very few notations to clearly illustrate the whole process, especially some are needed to be presented in formulas. **Since our work is the first attempt trying to do this, we believe a rigorous symbol system is necessray**. So, we clearly explain the main notations in Table 1 of Section B. We find that solely reducing notations are somewhat inappropriate. Oversimplified notations may introduce ambiguity, since notations can be simply removed, but the underlying concepts, operations, and our strategies still exist and cannot be discarded.
>
> **`Thanks again for your valuable comments. We promise we will try our best to polish our manuscript!`**
>
> &nbsp;
>
> ----
>
>
> ### **Q2. Question about SFS.**
> **A2.** Thanks for your good question! As we explain in the paper (line 330~335), **high-frequency information—though capturing finer visual information—tends to mislead VDMs to produce unnatural results, as it carries more noise perturbations [a, b]; by contrast, low-frequency information—while more robust—lacks sufficient fine-grained visual details.**
>
> The Switchable Frequency-domain Selection (SFS) aims to incorporate both high- and low-frequency information during drag-oriented latent optimization, while **`preventing the high-frequency component of latent features from dominating the drag process`** and avoiding the model **`overfitting`** to noise information in high frequency. This is achieved by **letting the model be randomly switchiable to different frequencies during iterative drag-based optimization** by randomly and uniformly selecting a cutoff frequency $\omega$ from {0.2, 0.4, 0.6, 1} in each iteration (*$\omega=1$ contains the full high-fequency component; a smaller $\omega$ represents latent features containing less high-frequency information*). Without SFS, the high-frequency componen of latent features persist throughout the latent optimization process; the model easily overfits to noisy information and produces artifacts and unnatural results, as shown in Figure 10 of Section F in the Appendix.
>
> >Reference:
> [a] Wavelet Integrated CNNs for Noise-Robust Image Classification;
> [b] Brief review of image denoising techniques.
>
> &nbsp;
>
> ----
>
>
> ### **Q3. Questions about window size in ADSR.**
> **A3.** Thanks for your constructive questions!
>
> 1. **How sensitive is the model to window size?** We want to highlight that **`our method is robust to the window size`**. The results are shown in the table below. By setting $window = 3, 9, 18$ frames, **the model performance is indeed stable and obviously higher than that of "w/o ADSR"**. The philosophy behind these results is very intuitive. We only need to **keep the latent distribution consistent before and after manipulation**. Despite the use of a small window size (e.g., 3), $window \times H\times W=3\times 60\times 104=18720$ latent feature vectors can still reflect the real distribution of latent embeddings, better than totally perturbed latent features in the "w/o ADSR" setting. Note that $H$ and $W$ denote the height and width of latent embeddings, respectively.
>
> &nbsp;
>
> | Window size | ObjMC ↓ |   DAI ↓    |   FVD ↓   |
> |:-----------:|:--------:|:--------:|:---------:|
> | w/o ADSR    | 45.09    | 0.093   | 608.63    |
> | 3           | 25.88    | 0.052   | 570.41    |
> | 9 (ours)    | 23.05    | 0.051   | 552.39    |
> | 18          | 24.37    |   0.051    | 554.25  |
>
> &nbsp;
>
> 2. **How is the window size determined?** As shown in the above table, our method is robust to the window size. **So, we just set it as a median value $window=9$.**
>
> 3. **Is there a trade-off?** There indeed exists a trade-off, and we **`agree`** with your opinion! However, as shown in the above table, this does not severely affect the final performance. **Using ADSR significantly outperforms removing it (“w/o ADSR”).**

---

> ### Author Response · Authors · 2025-11-28
> **Looking forward to receiving further feedback**
>
> *Dear Reviewer HUjr,*
>
> &nbsp;
>
>
> **We sincerely thank you again for your constructive comments—they truly mean a great deal to us!** We have carefully addressed your major concerns with detailed responses, analyses, and additional experiments. We have also uploaded a revised version of our paper, with all modifications clearly highlighted.
>
> Your insights have been invaluable in helping us strengthen our work! **We would greatly appreciate it if you could let us know whether there are any remaining issues or points that may require further clarification. We would be more than happy to provide any additional explanation you may need!**
>
> Thank you again for your time and valuable feedback. We look forward to hearing from you!
>
> &nbsp;
>
>
> *Best regards,*.
> *Authors of paper #1676*

---

### Official Review · Reviewer_XFrK · 2025-11-01

**Soundness:** 4
**Presentation:** 4
**Contribution:** 3
**Rating:** 6
**Confidence:** 3

**Summary:**

This article introduces a novel task, REVEL, and a corresponding solution, DragStream.

REVEL focuses on streaming-based video generation, enabling users to interactively modify any part of the frame at any time through dragging.

DragStream is a training-free approach that consists of two key modules: (1) ADSR, which addresses latent distribution drift, and (2) SFSO, which mitigates the dominance of high-frequency details in the context during the drag process.

Extensive quantitative and qualitative experiments demonstrate the effectiveness of the proposed method.

**Strengths:**

- This paper is written clearly and is easy to follow.
- It introduces a new task, REVEL, which focuses on streaming-based video generation, allowing users to interactively modify any part of the frame at any time through dragging.
- To address REVEL, the paper proposes a training-free method called DragStream, which consists of two core modules: ADSR and SFSO.
- ADSR addresses the issue of latent distribution drift by normalizing the current latents through statistical analysis of the mean and variance of neighboring latents, thereby adjusting the distribution.
- SFSO tackles the influence of high-frequency details in the context on the dragging process. Specifically, it uses Switchable Frequency-domain Selection (SFS) to control the amount of high-frequency information and Criticality-driven Spatial-domain Selection (CSS) to focus modifications on the handle region, reducing the impact on the global background.
- Quantitative and qualitative ablation experiments demonstrate the effectiveness of both ADSR and SFSO.
- Section D further discusses the differences between streaming/autoregressive models and non-streaming models.

**Weaknesses:**

- The efficiency of iterative latent region optimization (Sec. C.1) is stated with $I=4$, but I would like to know the runtime or complexity analysis. Since streaming-based generation is often used for real-time applications, can DragStream maintain sufficiently fast performance?
- For long video generation (e.g., exceeding 5 seconds, requiring multiple extensions), errors from autoregressive generation tend to accumulate over time. Is DragStream still effective in such cases where longer durations are involved?
- Does the denoising timestep ($T^′$) have a significant impact on DragStream? Is the choice of ($T^′$) dependent on the specific (distilled) video model?
- What happens if there is a conflict between the text prompt and the drag signal? For instance, if the text prompt directs a car to turn right while the drag signal indicates a turn to the left.
- Although Section D discusses non-streaming scenarios, It may also be worth validating DragStream on bi-directional models such as Wan2.1/2.2. If it proves effective in such cases, the training-free approach would be highly accessible to the community. At the very least, a discussion on the design considerations for such models would be valuable.
- My understanding may be incomplete, but it seems there could be a contradiction between on-the-fly editing and streaming generation. Typically, users might prefer to view the generated results first, then decide to modify a specific frame if it does not meet their expectations. However, in cases where the text prompt is incapable of achieving the desired outcome, users might apply drag operations before seeing the result. Could the authors elaborate more on the user cases of REVEL?

**Minor point**: The terms "ADSR" and "SFSO" could potentially be replaced with more intuitive, semantic meaningful names.

**Questions:**

I think the logic and contribution of this paper is clear. What I am most curious about are three points: the efficiency of DragStream, its performance in long video generation, and which hyperparameters are robust versus which are sensitive.

---

> ### Author Response · Authors · 2025-11-21
> **Response to Reviewer XFrK [1/3]**
>
> ### **Q1. Iterative optimization efficiency.**
> **A1.** Thanks for your good question! In fact, DragStream is **`sufficiently fast while also being friendly in computational resources`(requiring only one GPU card)**. Runtime analysis is provided in the table below. All the experiments are conducted on an H20 GPU, and runtime, ObjMC, and DAI are averaged over 204 video clips.
>
> &nbsp;
>
> | Experiments                 | Runtime/frame ↓ | ObjMC   ↓| DAI  ↓ |
> |:----------------------------:|:---------------:|:-----------:|:--------:|
> | $I = 4$                    | 0.30s          | 23.05     | 0.051  |
> | $I = 3$                    | 0.27s          | 24.55     | 0.053  |
> | $I = 2$                    | 0.24s          | 27.67     | 0.054  |
> | $I = 0$ (w/o our method) | 0.17s          | 90.39     | 0.133  |
>
> &nbsp;
>
> We do not carefully tune the hyperparameter $I$, as $I=4$ already yields satisfactory performance. Smaller values are also acceptable (e.g., $I=3$ or $2$), providing faster speed while still being substantially better than not using our approach ($I=0$). The above results domenstrate that the efficiency of our approach is indeed acceptable. **Kindly note that `H20 is an outdated GPU type`; using an A100 or H200 would yield even faster speed and higher efficiency.**
>
> &nbsp;
>
> ----
> ### **Q2. Compatibility with long video scenario (excceding 5s).**  
> **A2.** Thanks for your good question! Thanks to ADSR, which effectively suppresses the distribution drift of latent features, our method remains robust and effective in the long video scenario. **`For better displaying videos, the results are provided on Section I of our anonymous` [webpage](https://anonymous-30f12b51.github.io/Streaming-Drag-Oriented-Interactive-Video-Manipulation.github.io/ ).**   On one hand, **our method effectively supports drag-based manipulation in 5s, 10s, and 20s timestamps, despite autoregressively accumulated errors**; on the other hand, **drag operations induced by our approach indee do not noticeably degrade the quality of the subsequently generated video frames compared with the original videos generated by the baseline.**

---

> ### Author Response · Authors · 2025-11-21
> **Response to Reviewer XFrK [2/3]**
>
> ### **Q3. The hyperparameter $T'$ and robustness to hyperparameters.**
> **A3.** Thanks for your insightful questions!
> #### **1 . The hyperparameter $T'$.**
> - $T'$ indeed influences final results—also revealed by other prior visual editing works [a,b,c]—**since clean latent code is less editable whereas fully noise latent code loses too much visual content and cannot preserve high fidelity**. Following [a,b,c], we set $T'=\frac{3}{4}*T$, where $T$ is the number of total denoising timesteps.
>
> - $T'$ depends on noise level, **`but is model-agnostic!`** For CausVid, Self-Forcing, and even Wan2.1 (the experiment required by your Q5), we consistently set $T'=\frac{3}{4}*T$, in which latent code is editable (**not fully clean**) and retains essential original visual information (**not fully noisy**).
>
> >Reference:
> [a] Uncovering the Disentanglement Capability in Text-to-Image Diffusion Models, CVPR'23.
> [b] SG-I2V: Self-Guided Trajectory Control in Image-to-Video Generation, ICLR'25.
> [c] DragVideo: Interactive Drag-style Video Editing, ECCV'24.
>
> &nbsp;
>
> #### **2. Robustness to hyperparameters.**
> We would like to point out that **`hyperparameters are unavoidable`** in modern deep learning techniques and models. But **`our method is robust to the main hyperparameters in the approach`**, such as the iteration number $I$, the cutoff frequency $\omega$, and the shape of Gaussian filtering map $\alpha$.
>
> **Let us unpack the experimental results step by step to show the robustness of our DragStream!**
>
> - **Cutoff frequencies**. During latent region optimization, the cutoff frequency $\omega$ is randomly selected from the predefined cutoff frequency set {0.2, 0.4, 0.6, 1}; *$\omega=1$ denotes original latent embeddings that include **full high-frequency information***, and *$\omega=0.2$, $0.4$, or $0.6$ represents that only the **low-frequency component** of latent features is retained*. **`Our method is indeed robust to cutoff frequencies`**. As shown in the table below, despite letting the model randomly select low-frequency cutoff values from **`the continuous range`** [0.1, 0.8] in each optimiztion iteration, our method still yields stable performance.
>
> &nbsp;
>
> | Cutoff frequencies | ObjMC ↓ | DAI  ↓ | FVD ↓|
> |:------------------:|:-----:|:------:|:------:|
> | 0.2, 0.4, 0.6, 1    | 23.05 | 0.051 | 552.39 |
> | [0.1, 0.8], 1       | 24.27 | 0.054  | 584.21 |
>
> &nbsp;
>
> - **Gaussian filtering map**. The shape of the Gaussian filtering map is determined by the hyperparameter $\alpha$. The experimental results demonstrate our method' robustness again, which can be found on the table below.
>
> &nbsp;
>
> | $\alpha$          | ObjMC ↓ |   DAI  ↓  |  FVD ↓ |
> |:-----------------:|:---------:|:--------:|:-------:|
> | 2                 | 24.70   | 0.054   | 24.52   |
> | 1.5               | 23.50   | 0.052   | 24.17   |
> | 1.0 (ours)        | 23.65   | 0.053   | 24.04   |
> | 0.5               | 24.52   | 0.055   | 23.65   |
> | w/o Gaussian filtering map  | 26.12 | 0.061   | 25.16   |
>
> &nbsp;
>
> - **Iteration number**. In the following table, we also present experiments on the number of iterations used in drag-oriented latent optimization. Even with only 2 iterations, our approach still yields stable performance, offering faster speed while remaining substantially better than not using our method.
>
> &nbsp;
>
> | Experiments                 | Runtime/frame ↓ | ObjMC   ↓| DAI  ↓ |
> |:----------------------------:|:---------------:|:-----------:|:--------:|
> | $I = 4$                    | 0.30s          | 23.05     | 0.051  |
> | $I = 3$                    | 0.27s          | 24.55     | 0.053  |
> | $I = 2$                    | 0.24s          | 27.67     | 0.054  |
> | $I = 0$ (w/o our method) | 0.17s          | 90.39     | 0.133  |

---

> ### Author Response · Authors · 2025-11-21
> **Response to Reviewer XFrK [3/3]**
>
> ### **Q4. Text prompt and drag conflict.**
> **A4.** Thanks for your insightful and interesting question! We would like to clarify that **the model follows drag instructions first**, if there exist conflicts between text prompts and drag oprtations, since our latent region optimization revises latent embeddings more explicitly. For clearly displaying videos, we show the results on Section III of our anonymous [webpage](https://anonymous-30f12b51.github.io/Streaming-Drag-Oriented-Interactive-Video-Manipulation.github.io/).
>
> &nbsp;
>
> ----
>
> ### **Q5. Compatibility with bidirectional models.**
> **A5.** Thanks for your constructive suggestions! **Sure, our method can absolutely be applied to `bidirectional models`; actually, this setting is even easier than streaming drag, as perturbations are not amplified autoregressively**.  **`For better displaying videos, the results are provided on Section II of our anonymous`  [webpage](https://anonymous-30f12b51.github.io/Streaming-Drag-Oriented-Interactive-Video-Manipulation.github.io/).** Discussions are given below.
>
> Our DragStream is also suitable for bidirectional models. There are slight differences between streaming and non-streaming scenarios. For autoregressive VDMs, users can **observe** outputs as they are generated; **once any results are unsatisfactory, they can `immediately` modify current frames and their latent embeddings** to alter the subsequent generation direction. By contrast, **bidirectional VDMs generate all video frames simultaneously**. So, users should first observe the **`whole`** video, and then apply our method to latent features to revise generated video frames. Obviously, the **`advantage`** of our streaming-style drag is that users can modify generated results immediately, **`without having to wait for the generation of the entire video clip.`**
>
> &nbsp;
>
> ----
>
> ### **Q6. Paradigm contradiction.**
> **A6.** Thanks for your valuable comments! In fact, there is **`no contradiction`** between on-the-fly editing and streaming-style generation. We indeed let users observe generated results first. Specifically, our approach considers both editing and animation:
> >1. For editing, users **`first observe the outputs`** of autoregressive VDMs; then, once an  unsatisfactory or undesired frame is produced, users can immediately modify it via dragging (streaming but **not future**).
> 2. For animation, users **`first observe a currently generated video frame`**; then, drag instructions given by users are applied to objects in the frame (e.g., turning left/right) to animate and produce a video clip based on **the existing frame**.
>
> We would like to clarify that animation does not aim at editing future frames; it aims to **`animate an existing frame`** according to user instructions, e.g., dragging the car to the left (*information from future frames is not required; only the current existing video frame and the user’s instructions are required*). **Importantly, the latent features of animated video frames are stored via KV cache as contextual information, and guide the generation of subsequent video frames.** Editing and animation are complement each other, as users may either want to revise existing frames or to animate a video clip via interactively dragging (e.g., users can drag to animate a car running from left to right, a man lifting his head upward, etc).

---

> ### Author Response · Authors · 2025-11-28
> **Looking forward to receiving further feedback**
>
> *Dear Reviewer XFrK,*
>
> &nbsp;
>
>
> **We sincerely thank you again for your constructive comments—they truly mean a great deal to us!** We have carefully addressed each of your comments with detailed responses, analyses, and additional experiments. We have also uploaded a revised version of our paper, with all modifications clearly highlighted.
>
> Your insights have been invaluable in helping us strengthen our work! **We would greatly appreciate it if you could let us know whether there are any remaining issues or points that may require further clarification. We would be more than happy to provide any additional explanation you may need!**
>
> Thank you again for your time and valuable feedback. We look forward to hearing from you!
>
> &nbsp;
>
>
> *Best regards,*.
> *Authors of paper #1676*

---

### Author Response · Authors · 2025-11-21
**Summary of Rebuttal**

*Dear Reviewers, ACs, and PCs,*

&nbsp;

We appreciate the reviewers for **`acknowledging the several strengths`** of our work. For example,

>1. **`Reviewer XFrK`: "written clearly and easy to follow", "paper's logic and contribution are clear".**
2. **`Reviewer HUjr`: "highly useful and interesting", "results are impressive", "supported by a strong set of quantitative experiments and a detailed ablation study";**
3. **`Reviewer gQWk`: "well-written", "a clear and meaningful new task", "conceptually elegant";**
4. **`Reviewer y2YG`: "interesting task proposed", " relatively comprehensive experiment results", "both ADSR and SFSO are well-structured".**

We sincerely appreciate all reviewers' comments. **But we also want to friendly note that several of the major reviewer concerns stem from `misunderstandings` of the paper**; we would like to clarify:

>1. We do not claim "**Edit** Anything, Anytime"; we claim "**Drag** Anything, Anytime". These are totally different.
2. Our DragStream **does not aim at only editing future frames**; instead, it aims to **edit and animate existing frames** via interactive drag instructions.
3. Our method indeed **lets users first observe results**, and then modify results immediately via dragging if generated video frames are unsatisfactory and undesired.

We provide the following **additional experiments** during the rebuttal:
>1. We demonstrate that DragStream is **`sufficiently fast` while also being friendly in computational resources (`requiring only one GPU card`)**. Even on a single H20 GPU (**an outdated GPU type**), it only introduces 0.07s~0.13s runtime/frame.
2. We demonstrate that our approach is **`robust` to hyperparameters, such as the cutoff frequencies, the Gaussian filtering map spread, and the iteration number.**
3. We demonstrate that our approach can be applied to the **bidirectional model**, e.g., Wan2.1.
4. We demonstrate that our DragStream can successfully handle **long-video scenarios**.

For more details, please kindly refer to our point-by-point responses to reviewers.

***Lastly, we truly appreciate the time and effort that all reviewers, ACs, and PCs have devoted to our submission!***

&nbsp;

*Best regards,*
*Authors of paper #1676*

---

### Author Response · Authors · 2025-11-24
**Summary of Revisions Based on Reviewer Comments**

*Dear Reviewers, ACs, and PCs,*

&nbsp;


Based on the first-round comments and requirements from the reviewers, we have revised our manuscript accordingly, including


the paper's main body:
>1. we include the **runtime analysis** in Section 5.2 (`Table 1`; *kindly refer to page 9*);
2. we discuss **object occlusion and re-emergence** in Section 5.3 (`Figure 8`; *kindly refer to page 10*);
3. we discuss **streaming drag in long video generation** in Section 5.3 (`Figure 9`; *kindly refer to page 10*);
4. we include **failure cases** in Section 5.4 (`Figure 10`; *kindly refer to page 10*).

the appendix:
>1. we include a new **Section M: Robustness Analysis of Hyperparameters** (`Table4, Table 5, Table 6, Table 7`; *kindly refer to page 23*);
2. we include a new **Section N: Results on Controlling Bidirectional Models** (`Figure 17`; *kindly refer to page 23*)
3. we include a new **Section O: Discussion of Prompt-Drag Conflicts** (`Figure 18`; *kindly refer to page 24*)
4. we include a new **Section P: Additional Results on Long Video Generation** (`Figure 19`; *kindly refer to page 24*)
5. we include a new **Section Q: Long-Duration Drag-Oriented Manipulation** (`Figure 20`; *kindly refer to page 25*)
6. we include a new **Section R: Object Leaving and Re-Entering** (`Figure 21`; *kindly refer to page 26*)

 **Again, we sincerely thank all reviewers, ACs, and PCs for their time and effort in our submission! We welcome any additional questions during the discussion period.**


&nbsp;

*Best regards,*
*Authors of paper #1676*

---

### Author Response · Authors · 2025-11-30
**Rebuttal Summary for Newly Assigned AC**

#### *Dear re-assigned AC,*

&nbsp;

We sincerely appreciate your tremendous efforts in handling the urgent OpenReview information-leak incident! To speed up your review process, we would like to provide some critical information about our submission!

&nbsp;

---

**Firstly, all the four reviewers consistently acknowledge the strengths and novelty of our work.**

>1. Reviewer `gQWk`: **"a clear and meaningful new task", "conceptually elegant", "well-written";**
2. Reviewer `HUjr`: **"highly useful and interesting", "results are impressive";**
3. Reviewer `XFrK`: **"paper's logic and contribution are clear", "written clearly and easy to follow";**
4. Reviewer `y2YG`: **"interesting task proposed", "both ADSR and SFSO are well-structured", "comprehensive experiment results".**

---

 **Secondly, the reviewers gQWk, HUjr, and XFrK** `consistently keep positive scores` **throughout both the first-round review and the rebuttal period,** as they highly recognized our paper's contributions and novelty,
>*e.g.,* paving the way for Streaming Drag-oriented Interactive Video Manipulation by *(1)* proposing the REVEL task, *(2)* unifying drag-oriented video manipulation, and *(3)* designing a new DragStream framework.

---

**Thirdly**, we would like to kindly note that the reviewer y2YG had some **misunderstandings** about our work during the first-round review. *After our clarification in the rebuttal*, **the reviewer y2YG clearly stated in the official comments that** `he/she decided to raise the final score` **and accordingly applauded our work and efforts!**

---

&nbsp;

In summary, **we would like to highlight again that** `all four reviewers consistently express positive attitudes` **toward our paper** after the rebuttal; we also have addressed all the questions and concerns from the reviewers!







&nbsp;

#### *Best regards,*
#### *Authors of paper #1676*

---

### Meta-Review · Area_Chair_NLGW · 2025-12-28

**Summary:**

This submission received **consistently positive initial scores**, with three reviewers rating the paper at or above the acceptance threshold (Reviewers XFrK, HUjr, gQWk: all 6) and one initially critical reviewer (Reviewer y2YG: 2). During the rebuttal and discussion phase, the major concerns raised by Reviewer y2YG were clarified, and the reviewer explicitly stated improved understanding and raised the final score.

Reviewers consistently recognized the paper’s clear novelty in task formulation, namely REVEL, which formalizes streaming drag-oriented interactive video manipulation, as well as the practical and elegant design of the proposed training-free DragStream framework. While several reviewers raised concerns regarding presentation density, hyperparameter choices, runtime overhead, and long-horizon behavior, most of these concerns were addressed through detailed rebuttal explanations, additional experiments. Given the overall strength of the contribution, the clarity achieved after rebuttal, and the positive reviewer consensus, my judgment as AC is to **Accept**.

**Reviewer Concerns:**

### Concerns effectively addressed by the rebuttal

* **Runtime efficiency and real-time feasibility**:
  Multiple reviewers questioned whether the iterative latent optimization would incur prohibitive overhead for streaming scenarios
  *(Reviewers XFrK, gQWk, y2YG)*.
  The authors added per-frame runtime analysis, GPU memory usage, and demonstrated that DragStream runs at 0.07–0.13s per frame on a single H20 GPU, convincingly establishing practical feasibility.

* **Long-video and accumulated error behavior**:
  Reviewers expressed concerns about autoregressive error accumulation in long videos
  *(Reviewers XFrK, gQWk)*.
  The rebuttal included long-duration (up to ~20s) experiments and qualitative results showing that ADSR effectively suppresses latent drift over extended sequences.

* **Hyperparameter sensitivity (cutoff frequencies, window size, iterations)**:
  Several reviewers asked for clarification and robustness analysis
  *(Reviewers XFrK, HUjr, gQWk)*.
  The authors provided extensive robustness experiments, demonstrating stable performance across wide parameter ranges.

* **Prompt–drag conflicts and user interaction paradigm**:
  Reviewer XFrK and y2YG questioned potential contradictions between text prompts and drag signals, as well as the realism of “anytime” interaction. The rebuttal clarified the priority of drag signals.

* **Task definition misunderstandings**:
  Reviewer y2YG initially raised strong concerns about task realism and overclaiming. After clarification distinguishing “drag” from semantic editing, explaining that DragStream operates on existing frames (not future-only frames), and adding failure cases and runtime analysis, the reviewer explicitly acknowledged improved clarity and raised the score.

### Minor concerns that remain outstanding (non-blocking)

* **Presentation density and notation complexity**:
  While rigorously defined, the notation remains heavy and may hinder accessibility for a broader audience
  *(Reviewers HUjr, y2YG)*.
  Further streamlining or additional intuitive diagrams could improve readability.

* **Theoretical grounding of heuristic components**:
  Some design choices (e.g., frequency selection, ADSR windowing) are empirically motivated rather than theoretically analyzed
  *(Reviewer gQWk)*.
  This does not undermine effectiveness but leaves room for deeper theoretical study.

**Reviewer Scores:**

* **Reviewer XFrK (initial: 6)** → **Likely unchanged**
  Strongly positive on novelty, clarity, and experimental validation; questions fully addressed.

* **Reviewer HUjr (initial: 6)** → **Likely unchanged**
  Main concern on presentation; technical contributions and experiments viewed positively.

* **Reviewer gQWk (initial: 6)** → **Likely unchanged**
  Positive assessment of task novelty and method elegance; runtime and robustness concerns addressed.

* **Reviewer y2YG (initial: 2)** → **Improved after rebuttal (2 → 6)**
  Initial misunderstandings about task definition and claims were clarified; reviewer explicitly acknowledged improved understanding and raised the score.

---

### Decision · Program_Chairs · 2026-01-26

Accept (Poster)